# Microglia-mediated demyelination protects against CD8+ T cell-driven axon degeneration in mice carrying PLP defects

Janos Groh [1,2] ✉, Tassnim Abdelwahab [1], Yogita Kattimani[1],
Michaela Hörner [1,3], Silke Loserth[1], Viktoria Gudi[4], Robert Adalbert[5,6,7],
Fabian Imdahl[8], Antoine-Emmanuel Saliba [8,9], Michael Coleman [5],
Martin Stangel[4,10], Mikael Simons [2,11,12] & Rudolf Martini [1] ✉

Axon degeneration and functional decline in myelin diseases are often attributed to loss of myelin but their relation is not fully understood. Perturbed myelinating glia can instigate chronic neuroinflammation and contribute to demyelination and axonal damage. Here we study mice with distinct defects in the proteolipid protein 1 gene that develop axonal damage which is driven by cytotoxic T cells targeting myelinating oligodendrocytes. We show that persistent ensheathment with perturbed myelin poses a risk for axon degeneration, neuron loss, and behavioral decline. We demonstrate that CD8+ T cell-driven axonal damage is less likely to progress towards degeneration when axons are efficiently demyelinated by activated microglia. Mechanistically, we show that cytotoxic T cell effector molecules induce cytoskeletal alterations within myelinating glia and aberrant actomyosin constriction of axons at paranodal domains. Our study identifies detrimental axon-glia-immune interactions which promote neurodegeneration and possible therapeutic targets for disorders associated with myelin defects and neuroinflammation.

Myelination of axons by oligodendrocytes in the central nervous system (CNS) enables saltatory conduction, provides metabolic and trophic support, and modulates experience-dependent signal transmission[1,2]. However, these evolutionary benefits are associated with an increased vulnerability of white matter to various pathogenic processes including disease- and aging-related immune reactions[3–8]. Dysfunctional interactions between neural and immune cells are increasingly recognized to initiate and perpetuate neuroinflammation, contributing to white matter pathology and neurological dysfunction[9–11]. Oligodendrocyte-lineage cells are immunocompetent glial cells and actively participate in these processes[8,12–17]. Accordingly, chronic neuroinflammation is known to modify disorders associated with myelin defects and neurodegeneration, such as multiple sclerosis and hereditary leukodystrophies but also aging-related diseases like Alzheimer's and Parkinson's[18–21]. Axon degeneration in such disorders is most often proposed to be a consequence of chronic myelin and oligodendrocyte loss and the increased vulnerability of denuded axons to a toxic microenvironment[22–24]. However, the relationship between

[1]Department of Neurology, Section of Developmental Neurobiology, University Hospital Würzburg, Würzburg, Germany. [2]Institute of Neuronal Cell Biology, Technical University Munich, Munich, Germany. [3]Department of Neurology, Section of Neurodegeneration, University Hospital Heidelberg, Heidelberg, Germany. [4]Department of Neurology, Hannover Medical School, Hannover, Germany. [5]John van Geest Centre for Brain Repair, University of Cambridge, Cambridge, UK. [6]Department of Anatomy, Histology and Embryology, University of Szeged, Szeged, Hungary. [7]Institute of Health Sciences Education, Faculty of Medicine and Dentistry, Queen Mary University of London, London, UK. [8]Helmholtz Institute for RNA-based Infection Research, Helmholtz-Center for Infection Research, Würzburg, Germany. [9]Institute of Molecular Infection Biology (IMIB), University of Würzburg, Würzburg, Germany. [10]Translational Medicine, Novartis Institute of Biomedical Research, Basel, Switzerland. [11]German Center for Neurodegenerative Diseases, Munich, Germany. [12]Munich Cluster of Systems Neurology, Munich, Germany. ✉e-mail: janos.groh@tum.de; rudolf.martini@mail.uni-wuerzburg.de

immune reactions, demyelination, axon degeneration, and clinical disease is unclear, and many observations indicate that loss of myelin itself does not correlate well with progressive neurodegeneration[24–27].

We have previously demonstrated that gene defects perturbing the major CNS myelin proteolipid protein (PLP) - implicated in leukodystrophy and multiple sclerosis - result in neuroinflammation which amplifies neural damage and represents a target for treatment strategies[19]. In mice overexpressing normal or carrying mutant PLP (PLPtg and PLPmut mice, respectively) cytotoxic CD8+ T cells accumulate in the white matter and contribute to axon and myelin damage[28,29]. After cognate T cell receptor (TCR) engagement within the CNS, these cells promote an impairment of axonal transport and formation of axonal spheroids at juxtaparanodal domains of the nodes of Ranvier[30–32]. Moreover, innate immune reactions by microglia within the white matter contribute to neuroinflammation and myelin degradation in both disease models[33,34]. Despite these commonalities, the exact mechanisms by which cytotoxic T cells drive damage of myelinated axons are unclear and several independent studies have indicated that neural-immune interactions might be associated with distinct disease outcomes in these mice. While PLPtg mice predominantly develop a demyelinating phenotype with periaxonal vacuole formation and some other axonopathic alterations of unknown clinical impact[28,35], PLPmut mice show chronic progressive neurodegeneration, clinical disease, and comparatively mild demyelination[29,32,36]. Therefore, these models of rare hereditary diseases caused by defects within the same myelin gene offer unique opportunities to study how axon degeneration, demyelination and behavioral deficits are related in chronic neuroinflammation, with important implications for more common disorders.

Here we directly compare the progression of clinical and histopathological characteristics between PLPmut and PLPtg mice. In contrast to the widely accepted model that demyelination leads to axon loss, we find an inverse relationship of axon degeneration and demyelination. We characterize neural-immune interactions in both models and identify shared and distinct glial reactions. Using experimental approaches to manipulate microglial myelin phagocytosis, we observe that persistent ensheathment with perturbed myelin is a risk factor for neuroinflammation-driven axon degeneration and subsequent neuron loss. Along these lines, we uncover a mechanism of how myelinated axons are constricted at the paranodal domains as a reaction to the glia-directed CD8+ T cell attack.

## Results

### Persisting ensheathment with perturbed myelin correlates with T cell-driven neurodegeneration

Mice carrying mutant (PLPmut) or overexpressing normal (PLPtg) PLP have been reported to display similarities and differences in pathogenesis and disease outcome (Table 1). Due to these characteristics, we set out to address the following questions in the disease models: How

are axon degeneration and demyelination related to disability and to each other? How are overlapping but distinct immune reactions related to axon degeneration and demyelination? How do glia-directed immune reactions drive axonal damage and degeneration? To characterize the functional impact of the distinct perturbations affecting myelinating glia we used independent behavioral readout measures representing different neural circuits. While PLPmut mice (R137W, homozygous[29]) showed a significant decline in rotarod performance compared with non-transgenic wildtype (Wt) mice between 12 and 18 months of age, PLPtg mice (line 66, hemizygous[37]) retained normal motor coordination (Fig. 1a). Similarly, PLPmut but not PLPtg mice demonstrated a progressive decline in visual acuity that started even before 12 months of age (Fig. 1b). Structurally, this was accompanied by a more pronounced loss of retinal ganglion cells in PLPmut compared with PLPtg mice (Fig. 1c, d). Optical coherence tomography (OCT) also demonstrated earlier and more prominent thinning of the innermost retinal layers in PLPmut mice (Fig. 1e, Supplementary Fig. 1a), arguing for more severe neuroaxonal degeneration. Electron microscopic quantification confirmed more axonopathic profiles (Supplementary Fig. 1b) and a more severe loss of axons in optic nerves of PLPmut compared with PLPtg mice at 12 months of age (Fig. 1f, g). In contrast, the frequency of thinly or non-myelinated axons was significantly higher in PLPtg compared with PLPmut mice (Fig. 1f,h). However, both disease models showed similar dynamics regarding the formation of axonal spheroids, which we labelled using antibodies against non-phosphorylated neurofilaments (SMI32, Supplementary Fig. 1c). Axonal spheroid formation was associated with an accumulation of CD8+ T cells in the white matter parenchyma of both models (Supplementary Fig. 1d–f), in line with their previously identified causal involvement in axonal damage[28,29,31,32]. In summary, we detected an inverse relationship between neurodegeneration (associated with functional decline) and demyelination when comparing the two mouse models, despite an initially similar T cell-driven damage of myelinated axons.

We then compared the formation and fate of axonal spheroids in more detail. Electron microscopy and immunohistochemistry indicated that axonal spheroids in PLPmut mice were more often of larger size and still myelinated, while axonal spheroids appeared smaller and frequently non-myelinated in PLPtg mice (Fig. 2a, b). Detailed analyses at multiple ages revealed that axonal spheroids with small diameter (1.5 to 4 μm) form early in both models and are detectable at all investigated ages (Fig. 2c). However, axonal spheroids frequently progressed to larger sizes (>4 μm diameter) in PLPmut mice, especially at advanced disease stages (12 and 18 months of age) while this was not the case in PLPtg mice. Additionally, we observed that axonal spheroids initially are still mostly myelinated in both models and become demyelinated with disease progression in PLPtg but rarely in PLPmut mice (Fig. 2d, e, Supplementary Fig. 1e). Thus, neuroinflammation-related axonal spheroids initially form similarly within perturbed

**Table 1 | Similarities and differences in pathogenesis and disease outcome of mice carrying mutant (PLPmut) compared with mice overexpressing normal (PLPtg) PLP**

| Disease model | PLPmut (homozygous) | PLPtg (hemizygous) |
|---|---|---|
| Primary defect | CNS myelin protein (PLP1 point mutation[29]) | CNS myelin protein (Plp1 overexpression[37]) |
| Cellular players of neuroinflammation | Microglial activation, cytotoxic CD8+ T cells[29,32,34] | Microglial activation, cytotoxic CD8+ T cells[28,30,31] |
| Impact of neuroinflammation | Axonal transport defects and spheroid formation[29,32,34] | Axonal transport defects and spheroid formation[28,30,31] |
| | Mild demyelination[29,32,34] | Prominent demyelination[28,33] |
| | Visual and motor dysfunction[29,32,36] | Preservation of visual and motor function (this study) |
| | Prominent neuron loss[29] | Mild neuron loss (this study) |
| | Large axonal spheroids and severe axon loss[29,32,34] | Small axonal spheroids and mild axon loss (this study) |
| | Modest microglial myelin phagocytosis (this study) | Prominent microglial myelin phagocytosis (this study) |

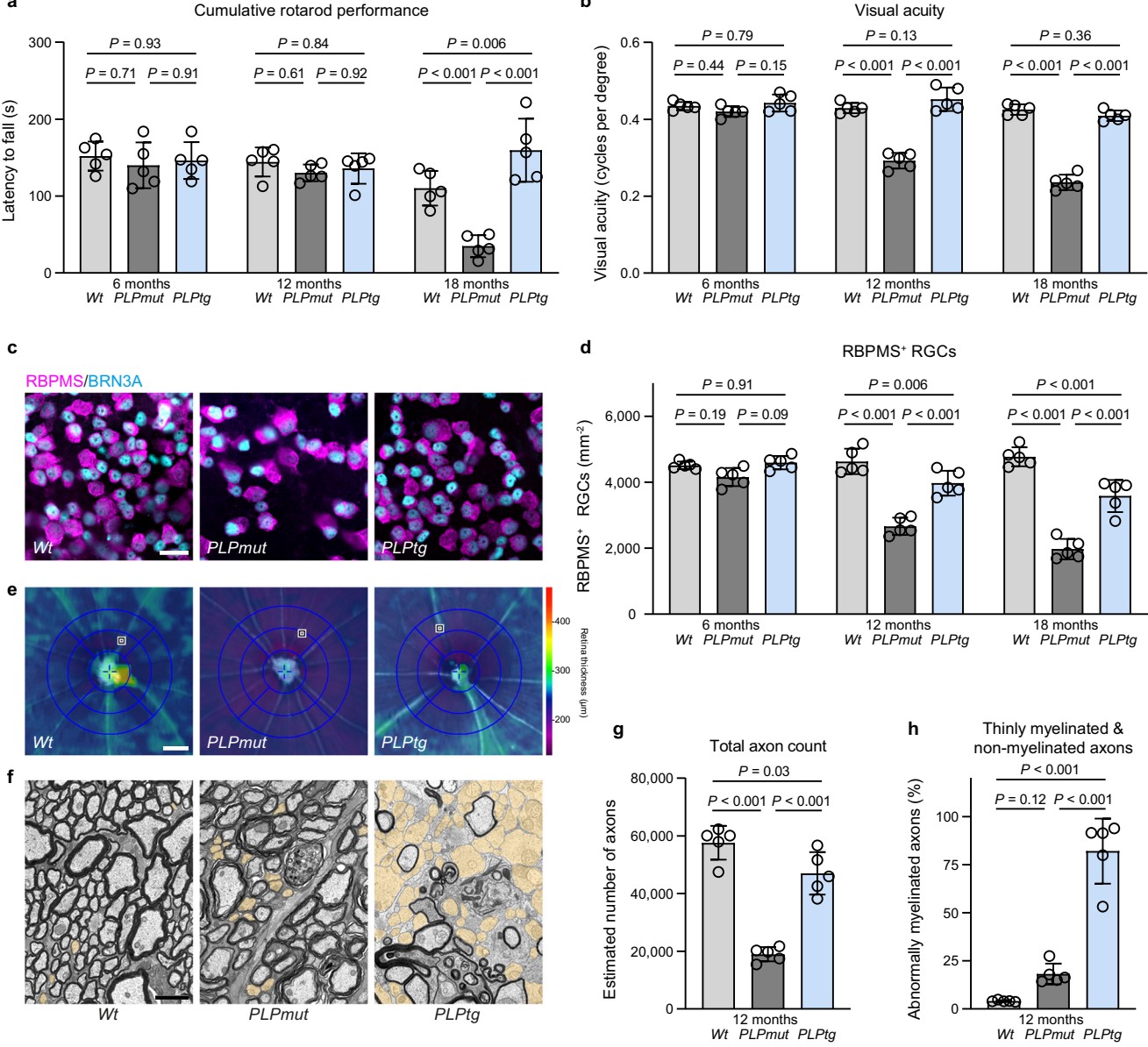

**Fig. 1 | Neurodegeneration and myelin loss correlate inversely in mice with distinct myelin defects. a** Accelerating rotarod analysis of motor performance in *Wt*, *PLPmut*, and *PLPtg* mice (each circle represents the mean value of five consecutive runs of one mouse) at different ages. Motor performance is significantly impaired in *PLPmut* but not *PLPtg* mice at advanced disease stage (*n* = 5 mice per group, two-way ANOVA with Tukey's multiple comparisons test, *F* (2, 36) = 14.88, *P* < 0.001). **b** Automated optokinetic response analysis of visual acuity (cycles per degree) shows a progressive decline of visual acuity in *PLPmut* but not *PLPtg* mice (each circle represents the mean value of one mouse, *n* = 5 mice per group, two-way ANOVA with Tukey's multiple comparisons test, *F* (2, 36) = 195.0, *P* = 0.001). **c** Immunofluorescence detection and (**d**) quantification of RBPMS⁺BRN3A⁺ RGCs in the retinae of *Wt*, *PLPmut*, and *PLPtg* mice (*n* = 5) mice per group, two-way ANOVA

with Tukey's multiple comparisons test, *F* (2, 36) = 112.3, *P* < 0.001). Scale bar, 20 μm. **e** Representative interpolated thickness maps from retinal volume scans of 12-month-old *Wt*, *PLPmut*, and *PLPtg* mice. Scale bar corresponds to 50 μm of subtended retina. **f** Representative electron micrographs of optic nerve cross-sections from 12-month-old *Wt*, *PLPmut*, and *PLPtg* mice (thinly and non-myelinated axons are indicated in yellow pseudocolor). Scale bar, 2 μm. **g** Electron microscopy-based estimation of total axonal numbers in the optic nerves. Axon loss is much milder in *PLPtg* than in *PLPmut* mice (*n* = 5 mice per group, one-way ANOVA with Tukey's multiple comparisons test, *F* (2, 12) = 63.67, *P* < 0.001). **h** Quantification of thinly myelinated (g-ratio ≥ 0.85) and non-myelinated axons in *Wt*, *PLPmut*, and *PLPtg* mice (*n* = 5 mice per group, one-way ANOVA with Tukey's multiple comparisons test, *F* (2, 12) = 81.47, *P* < 0.001). Data are presented as the mean ± s.d.

myelin segments, but they only appear to progress in size and ultimately lead to axon degeneration when they remain myelinated. In line with these observations, CD8⁺ T cells in *PLPtg* mice preferentially associated with SMI32⁺ fibres that were still myelinated at an age when around 80% of optic nerve axons are demyelinated (Supplementary Fig. 2). These observations argue against the view that loss of myelin makes axons susceptible to secondary immune-mediated degeneration and instead imply that efficient demyelination may allow resilience of axons at reversible stages of damage.

## Distinct interactions between perturbed oligodendrocytes and microglia determine axonal demyelination

To better characterize the cellular interactions that determine these different outcomes regarding the persistence of perturbed myelin, we investigated transcriptional changes within myelinating glia and CNS-resident phagocytes. We therefore used single-cell RNA sequencing (scRNA-seq) of CD45⁻O1⁺ mature oligodendrocytes and CD45^low^Siglec-H⁺ microglia sorted from the brains of adult (10-month-old) *Wt*, *PLPmut*, and *PLPtg* mice (gating strategy presented in Supplementary

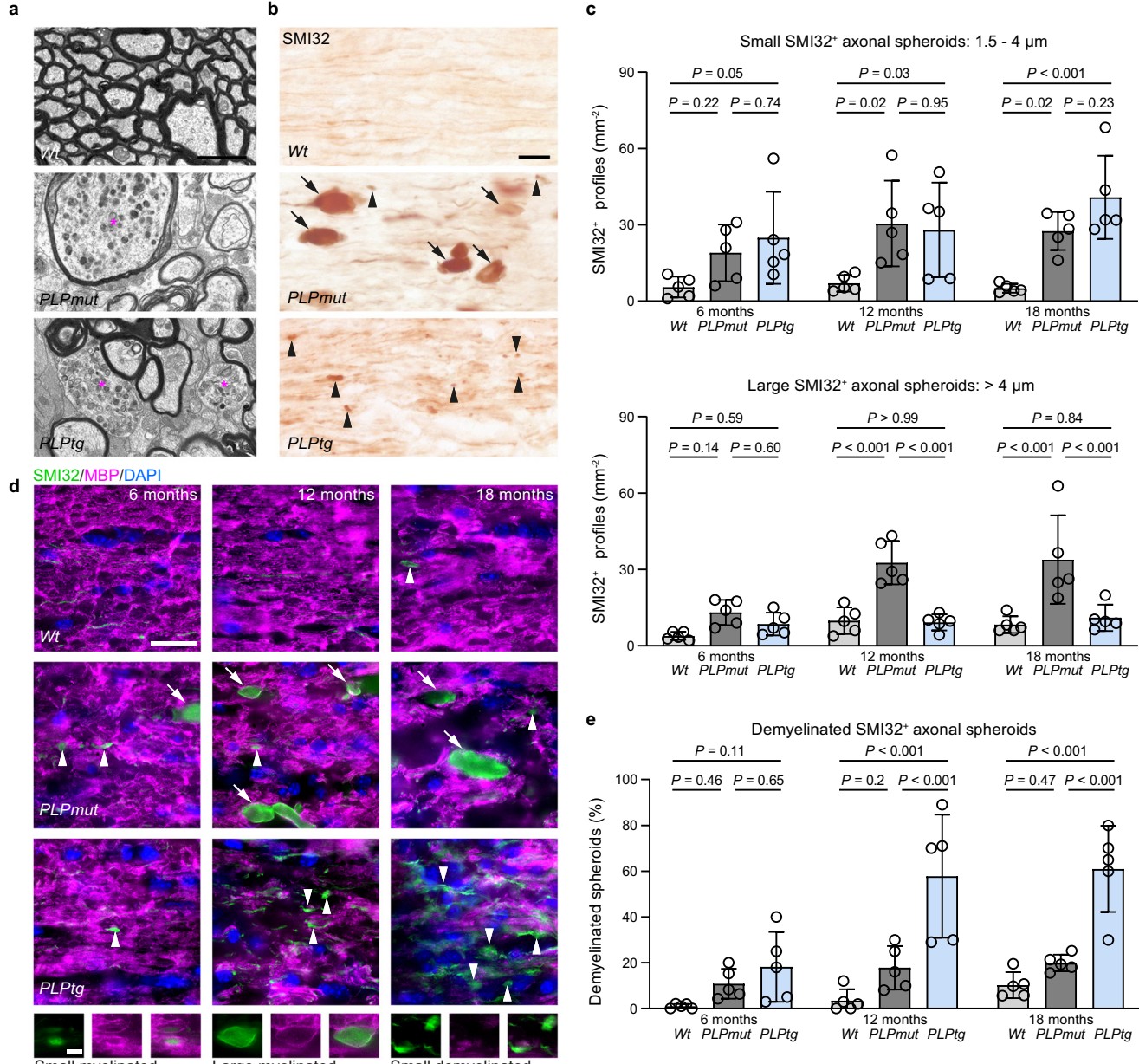

**Fig. 2 | Axonal spheroids show a similar initial formation but a different progression in mice with distinct PLP defects. a** Representative electron micrographs of optic nerve cross-sections from 12-month-old *Wt*, *PLPmut*, and *PLPtg* mice demonstrate differences in size and myelination state of axonal spheroids (asterisks). Scale bar, 2 μm. **b** Immunohistochemical visualization of SMI32⁺ axonal spheroids in the optic nerves of 12-month-old *Wt*, *PLPmut*, and *PLPtg* mice. Arrowheads indicate small axonal spheroids (diameter 1.5–4 μm) and arrows indicate large axonal spheroids (diameter >4 μm). Scale bar, 20 μm. **c** Quantification of small (top) and large (bottom) axonal spheroids at different ages (each circle represents the mean value of one mouse). Small spheroids are similarly frequent in *PLPmut* and *PLPtg* mice but become large with disease progression only in *PLPmut*

mice ($n = 5$ mice per group, two-way ANOVA with Tukey's multiple comparisons test, Small: $F_{(2, 36)} = 16.71$, $P < 0.001$, Large: $F_{(2, 36)} = 30.04$, $P < 0.001$).
**d** Immunofluorescence detection of SMI32⁺ axonal spheroids (small: arrowheads, large: arrows) and MBP in the optic nerves of *Wt*, *PLPmut*, and *PLPtg* mice at different ages. Scale bar, 20 μm. Bottom images show examples of spheroids of different sizes and myelination states with split channels at higher magnification. Scale bar, 5 μm. **e** Quantification of the myelination state of SMI32⁺ axonal spheroids demonstrates their progressive demyelination in *PLPtg* but not *PLPmut* mice ($n = 5$ mice per group, two-way ANOVA with Tukey's multiple comparisons test, $F_{(2, 36)} = 39.45$, $P < 0.001$). Data are presented as the mean ± s.d.

Fig. 3a). After quality control and filtering (Supplementary Fig. 3b), unsupervised clustering of the combined datasets identified nine different microglia clusters and a smaller oligodendrocyte (ODC) cluster (Fig. 3a, Supplementary Data 1). Within ODC, we detected shared and unique disease-related transcriptional alterations in the distinct models. Both in *PLPmut* and *PLPtg* mice several transcripts indicating altered proteostasis and cell stress (e.g., *Uba52*, *Cd63*, *Rps5*), and inflammatory signalling (e.g., *Il33*, *Ccl4*, *Lyz2*) were differentially

expressed, likely as a direct consequence of the distinct myelin gene defects (Supplementary Fig. 3c, d, Supplementary Data 1). However, while ODC in *PLPmut* mice upregulated several transcripts related to protection against cell death and phagocytosis (e.g., *Cryab*, *Xiap*, *Cd47*), ODC in *PLPtg* mice expressed high levels of phagocytosis-promoting transcripts (e.g., *C1qa*, *C1qb*, *Apoe*).

We next focused on the consequences of these alterations for microglial diversity and function. Four of the microglia clusters

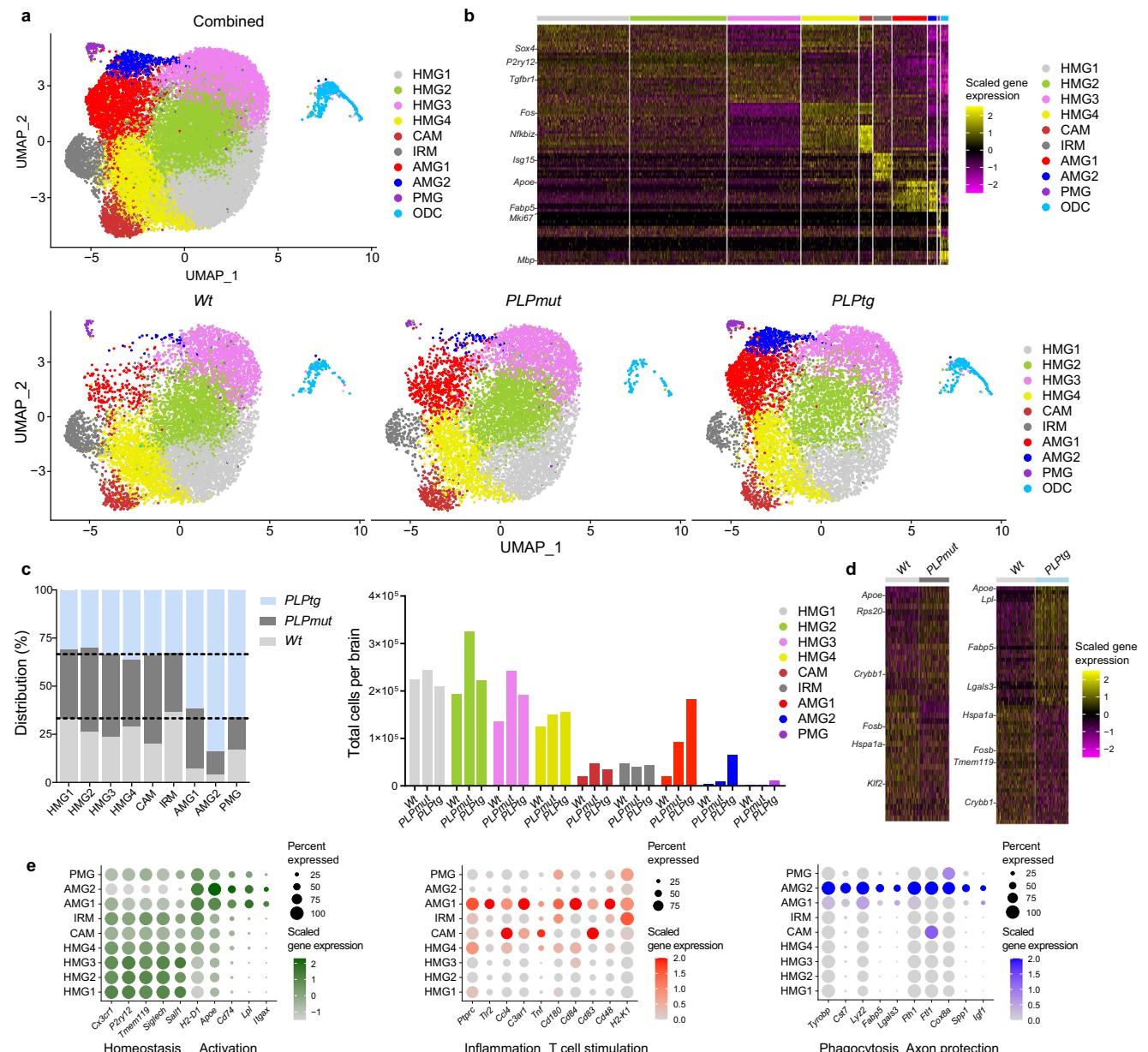

**Fig. 3 | scRNA-seq reveals heterogeneous neural-immune interactions in mice with distinct myelin defects. a** UMAP visualization of CD45−O1⁺ mature oligodendrocytes and CD45^low^Siglec-H⁺ microglia freshly sorted from adult (10-month-old) *Wt*, *PLPmut*, and *PLPtg* (*n* = 3 mice per group) mouse brains and analyzed by scRNA-seq. Combined (top, 26,308 cells) and separate (bottom) visualization of cells from *Wt* (9,484 cells), *PLPmut* (8,405 cells), and *PLPtg* (8,419 cells) brains are displayed. **b** Heatmap of top 10 cluster-specific genes. The colour scale is based on a z-score distribution from −2 (purple) to 2 (yellow). **c** Contribution of the samples to each microglia cluster is displayed in percent (left) and absolute numbers extrapolated to total cells per brain (right). AMG1 is enriched in both myelin mutants and AMG2 mostly in *PLPtg* mice. **d** Heatmaps of top 30 differentially expressed genes

comparing microglia isolated from *Wt* and *PLPmut* (left) or *Wt* and *PLPtg* (right) brains across all clusters as identified in panel (**a**). **e** Dot plot expression visualization of selected genes implicated in microglial homeostasis and activation (left), inflammation and T cell stimulation (middle), or phagocytosis and axon protection (right) for microglia clusters as annotated in panel a. The color scales are based on z-score distributions from −1 (lightgrey) to 2 (green) or 0 (lightgrey) to 2 (red, blue). HMG, homeostatic microglia; CAM, capillary-associated microglia; IRM, interferon-responsive microglia; AMG, activated microglia; PMG, proliferating microglia; ODC, oligodendrocytes. Complete lists of cluster-specific markers and differentially expressed genes can be found in Supplementary Data 1.

resembled homeostatic microglia (HMG1–4; Fig. 3b)[38]. Moreover, we identified one cluster with a signature enriched in immediate early genes and *Icam1* and one cluster with a prominent interferon response signature, reminiscent of previously described capillary-associated microglia (CAM, see below) and interferon-responsive microglia (IRM), respectively[39,40]. We also detected the presence of two groups of activated microglia (AMG1-2) with partially overlapping transcriptional signatures and a small population of proliferating microglia (PMG). Both AMG clusters showed transcriptional similarities to previously

described disease-, aging-, or neuroinflammation-related microglia states such as DAM[41], MGnD[42], WAM[43], microglia in EAE[44], after systemic LPS challenge[45], and to a smaller extent DIM[46] (which showed more transcriptional similarities to CAM; Supplementary Fig. 4a, Supplementary Data 1). Moreover, AMG and PMG were strongly enriched in the disease models, with AMG2 and PMG being more unique to *PLPtg* mice (Fig. 3c). Global myelin disease-related transcriptional changes in microglia were primarily based on increased numbers of cells representing AMG and reflected the prominent contribution of

AMG2 in *PLPtg* mice (Fig. 3d). Focusing on the activated microglia states, we found that both shared (to different degrees) the typical downregulation of markers indicating homeostatic function (e.g., *Cx3cr1, P2ry12, Tmem119, Siglech, Sall1*) and the upregulation of activation markers (e.g., *H2-D1, Apoe, Cd74, Lpl, Itgax*) reminiscent of the DAM program[47] (Fig. 3e, Supplementary Fig. 4b). However, when looking for subset-enriched signatures (marker genes that are also significantly upregulated in one vs the other AMG cluster), we observed that AMG1 expressed higher levels of genes associated with pro-inflammatory signaling (e.g., *Ptprc, Tlr2, Ccl4, C3ar1, Tnf*) and T cell stimulation (e.g., *Cd180, Cd84, Cd83, Cd48, H2-K1*), whereas AMG2 showed enrichment of genes related to phagocytosis (e.g., *Tyrobp, Cst7, Lyz2, Fabp5, Lgals3*)[47,48] and axon protection (e.g., *Fth1, Ftl1, Cox8a, Spp1, Igf1*)[49,50]. GO analysis reflected these differences between AMG states (Supplementary Fig. 4c). Flow cytometry of CD45$^{low}$Siglec-H$^+$ microglia from brains of 12-month-old *Wt, PLPmut*, and *PLPtg* mice using identified markers confirmed the presence of distinct populations and increased frequency of the AMG subsets in both disease models (Supplementary Fig. 5). Using CD11c (encoded by *Itgax* and specific to both AMG clusters) and PD-1 (encoded by *Pdcd1* and specific to AMG1) as discriminating markers, we further confirmed the stronger accumulation of AMG2 in *PLPtg* mice.

Immunohistochemistry revealed that AMG (CD11c$^+$) specifically arise in the white matter of mice with myelin perturbation (Supplementary Fig. 6a, b). Moreover, CAM (ICAM1$^+$) were localized in proximity to capillaries in adult *Wt* mice (Supplementary Fig. 6c). Interferon-responsive microglia and astrocytes were previously localized to white matter regions in proximity to the ventricles[8,51,52]. Enzymatic dissociation artifacts[53] could not fully explain the detection of the distinct HMG subsets, since c-Jun expression levels (part of the ex vivo activation signature and differentially expressed between HMG subsets) varied between individual microglia in adult *Wt* mice (Supplementary Fig. 6d, e). In addition, MHC-I expression (differing between HMG1-2 and HMG3-4) was higher in *Wt* white matter compared with grey matter microglia (Supplementary Fig. 6d, f). Electron microscopy and immunohistochemistry showed that myelin-phagocytosing amoeboid microglia with an accumulation of lysosomal storage material were frequent in *PLPtg* but not *PLPmut* mice (Fig. 4a, b). Immunohistochemical quantification confirmed that the total number of microglia (CD11b$^+$) as well as AMG (CD11c$^+$) was increased in optic nerves of both disease models. However, while AMG1 (P2RY12$^+$CD11c$^+$) was enriched in both models, AMG2 (P2RY12$^-$ CD11c$^+$) was specific to *PLPtg* mice (Fig. 4c–e). In line with our scRNA-seq data, galectin 3 (encoded by *Lgals3*), associated with phagocytosis[48], was detectable at higher levels in AMG2 (P2RY12$^-$CD11c$^+$) of *PLPtg* mice than in AMG1 (P2RY12$^+$CD11c$^+$) of *PLPmut* or HMG (P2RY12$^+$CD11c$^-$) of *Wt* mice. In summary, distinct myelin defects within oligodendrocytes of *PLPmut* and *PLPtg* mice appear to drive similar pro-inflammatory microglial activation (represented by AMG1). However, they result in varying responses regarding the transition into a myelin-phagocytosing microglia state (predominantly represented by AMG2). These reactions might depend on cell-intrinsic differences related to the myelin gene defects or represent distinct responses to the glia-directed CD8$^+$ T cell attack.

### Microglia-mediated removal of perturbed myelin protects against CD8$^+$ T cell-driven axon degeneration

Our previous data suggested that persisting ensheathment with perturbed myelin makes axons susceptible to neuroinflammation-related damage and degeneration in distinct disease models. We therefore wanted to test the hypothesis that microglia-dependent demyelination can protect against axonal degeneration. Thus, we used two complementary approaches aiming to modulate demyelination. First, we fed *PLPmut* mice with the copper chelator cuprizone to enforce toxin-

based demyelination known to be dependent on CSF-1-activated microglia[54,55]. Second, we pharmacologically depleted microglia with the CSF-1R inhibitor PLX5622 to attenuate demyelination in *PLPtg* mice[56]. These approaches were selected to foster or mitigate the predicted impact of microglial myelin phagocytosis based on their observed activation states and our previous work in the respective models[33,34]. Electron microscopy-based quantification revealed a significant increase in thinly and non-myelinated axons in optic nerves of *PLPmut* mice after 6 weeks of cuprizone treatment (Fig. 5a). This was associated with fewer axons showing pathological alterations in form of spheroid formation and degenerative signs (Fig. 5b) and resulted in an attenuation of axon loss (Supplementary Fig. 7a). Moreover, cuprizone treatment increased the frequency of *Csf1*$^+$*Mbp*$^+$ oligodendrocytes and amoeboid microglia with lysosomal storage material in *PLPmut* mice (Supplementary Fig. 7b, c). On the other hand, 6 months of continuous PLX5622 treatment resulted in a reduced frequency of thinly- and non-myelinated axons and an increased frequency of axonal spheroids and degenerating axons in *PLPtg* but not *Wt* mice (Fig. 5c, d). This led to an aggravated loss of axons and retinal ganglion cells (Supplementary Fig. 7d, e). To study the consequences of even faster removal of perturbed myelin on axon integrity, we also investigated homozygous *PLPtg* mice. In contrast to hemizygous mice, these showed almost complete absence of myelin in the optic nerves at 4 months of age without detectable axon loss (Supplementary Fig. 7f, g). Corroborating our observations regarding myelin phagocytosis, cuprizone treatment led to a significant increase in the number of AMG2 (P2RY12$^-$CD11c$^+$) expressing galectin 3[48] (encoded by *Lgals3*) in *PLPmut* mice, while PLX5622 decreased their density in *PLPtg* mice (Fig. 5e, f). These observations indicate that fostering microglia-mediated demyelination attenuates axonal degeneration and that impairing demyelination aggravates axonal degeneration when perturbed myelin is targeted by adaptive immunity.

### Focal damage of axons ensheathed by perturbed myelin occurs in proximity to constricted paranodal regions

Next, we addressed the underlying mechanisms responsible for the formation of spheroids as an early sign of focal damage by CD8$^+$ T cells to axons ensheathed by perturbed myelin. Upon close examination of longitudinal optic nerve sections by electron microscopy, we confirmed the previously described preferential localization of spheroids at juxtaparanodal axon domains[29,31] and frequently observed an accumulation of disintegrating mitochondria and other organelles (an early sign of axonal damage[57,58]). Strikingly, at the corresponding paranodal domains, axons appeared to be of smaller diameter than usual (Fig. 6a). In *PLPmut* mice, progressive swelling, accumulation of dense bodies within myelinated spheroids, and fragmentation seemingly correlated with decreasing paranodal diameters. In *PLPtg* but not *PLPmut* mice, we frequently detected phagocytosing microglia stripping early-stage axonal spheroids of myelin. Abnormally small paranodal axon diameters were also detectable in cross sections of *PLPmut* optic nerves, excluding misinterpretation related to oblique fibre orientation in longitudinal sections (Fig. 6b). This previously unknown narrowing of the paranodal axon diameter reminiscent of a constriction process could explain the described impairment of axonal transport and consequent juxtaparanodal accumulation of organelles in the disease models[29,31]. We, therefore, measured the minimum axonal diameters of paranodal domains with adjacent spheroid formation categorized into different putative stages of progression (Fig. 6c). Normal appearing fibres in *Wt, PLPmut*, and *PLPtg* mice displayed larger paranodal axon diameters than those at stages 1 (accumulation of organelles) and 2 (swelling). An increased frequency especially of stage 2 profiles resulted in a significant reduction of the mean paranodal axon diameter in *PLPmut* compared with *Wt* and *PLPtg* mice. Moreover, in longitudinal sections, oligodendroglial cytoplasmic loops displayed decreased circularity going along with progressive spheroid

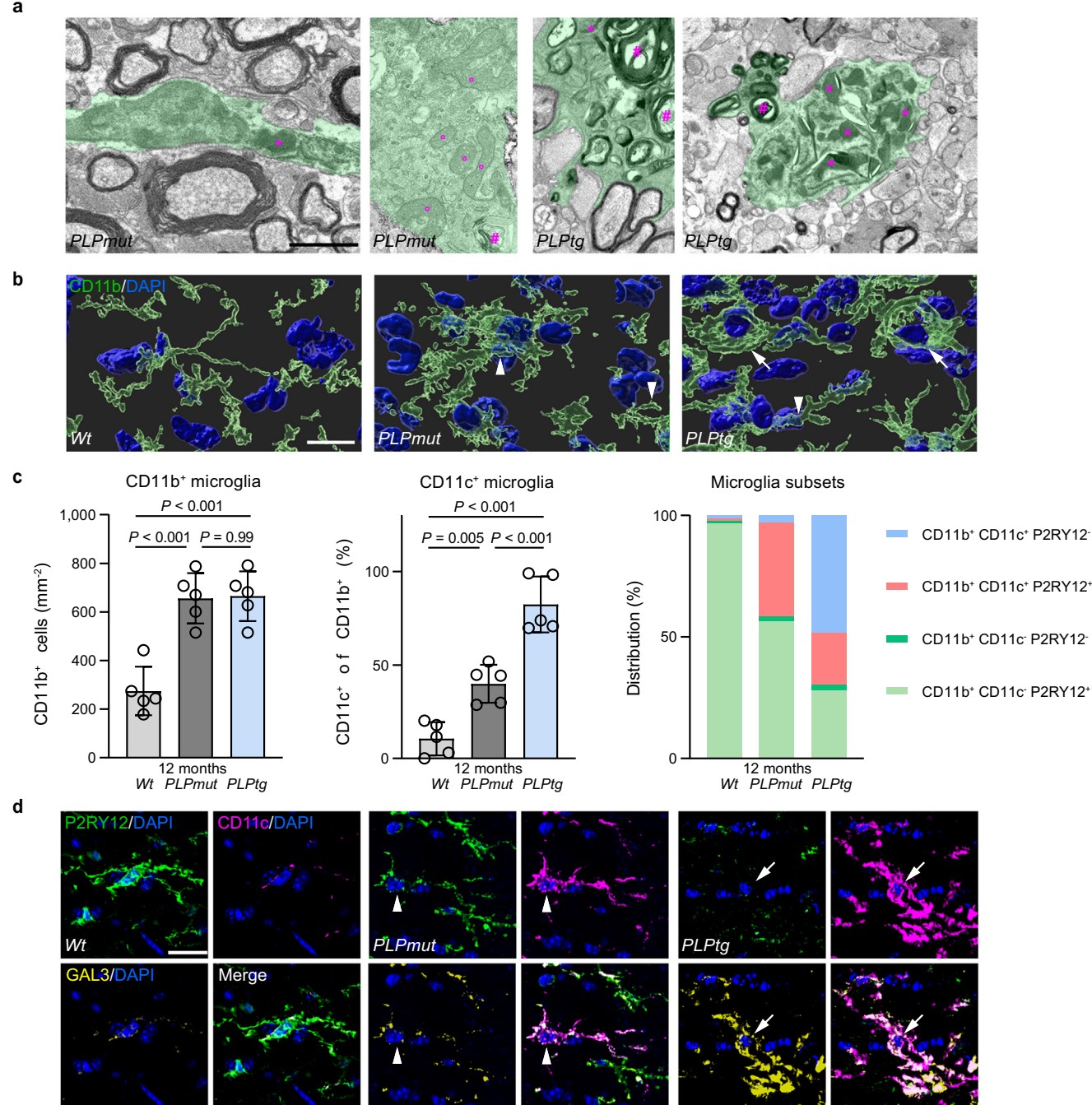

**Fig. 4 | Myelin phagocytosis by activated microglia mediates demyelination in mice with distinct PLP defects. a** Representative electron micrographs of microglial cells (green pseudocolor) in optic nerve cross-sections from 12-month-old *PLPmut*, and *PLPtg* mice demonstrate differences in morphology, mitochondrial content (circles), and intracellular accumulation of myelin fragments (hashtags) and lysosomal storage material (asterisks). Scale bar, 2 μm.
**b** Immunofluorescence detection and IMARIS Z-stack surface rendering of CD11b⁺ microglia in optic nerves of *Wt*, *PLPmut*, and *PLPtg* mice. Arrowheads indicate "bushy" microglia with thick processes and arrows indicate "amoeboid" microglia with short processes. Scale bar, 10 μm. **c** Quantification of CD11b⁺ microglia (left), CD11c⁺ microglia (% of CD11b, middle), and distribution of CD11c/P2RY12 reactivity on microglia (% of CD11b, right) in the optic nerves of 12-month-old *Wt*, *PLPmut*, and

*PLPtg* mice (each circle represents the mean value of one mouse) as shown in Supplementary Fig. 6a, b. Ramified CD11c⁺ P2RY12⁺ microglia (representing AMG1) accumulate in both myelin mutants while amoeboid CD11c⁺ P2RY12⁻ microglia (representing AMG2) arise in *PLPtg* mice ($n = 5$ mice per group, one-way ANOVA with Tukey's multiple comparisons test, Left: $F_{(2, 12)} = 23.88$, $P < 0.001$, Middle: $F_{(2, 12)} = 48.00$, $P < 0.001$). **d** Representative immunofluorescence detection of P2RY12, CD11c, and GAL3 in the optic nerves of 12-month-old *Wt*, *PLPmut*, and *PLPtg* mice. CD11c⁺ P2RY12⁻ (AMG2, arrow) microglia in *PLPtg* mice show higher expression of GAL3 than CD11c⁺ P2RY12⁻ (AMG1, arrowhead) microglia in *PLPmut* or CD11c⁻ P2RY12⁺ (HMG) microglia in *Wt* mice. Scale bar, 10 μm. Data are presented as the mean ± s.d.

formation (Supplementary Figs. 8a, 6d). This suggests that abnormal myelinating process extension may contribute to constriction of paranodal axon diameters. Focusing on *PLPmut* mice, we further studied the putative role of paranodal constriction in axon degeneration

and tested its dependency on functional adaptive immunity. *Rag1*-deficient *PLPmut* mice (lacking mature adaptive immune cells) showed less decreased paranodal axon diameters (Supplementary Figs. 8b, 6e). We confirmed this immune-dependent overall decrease in paranodal

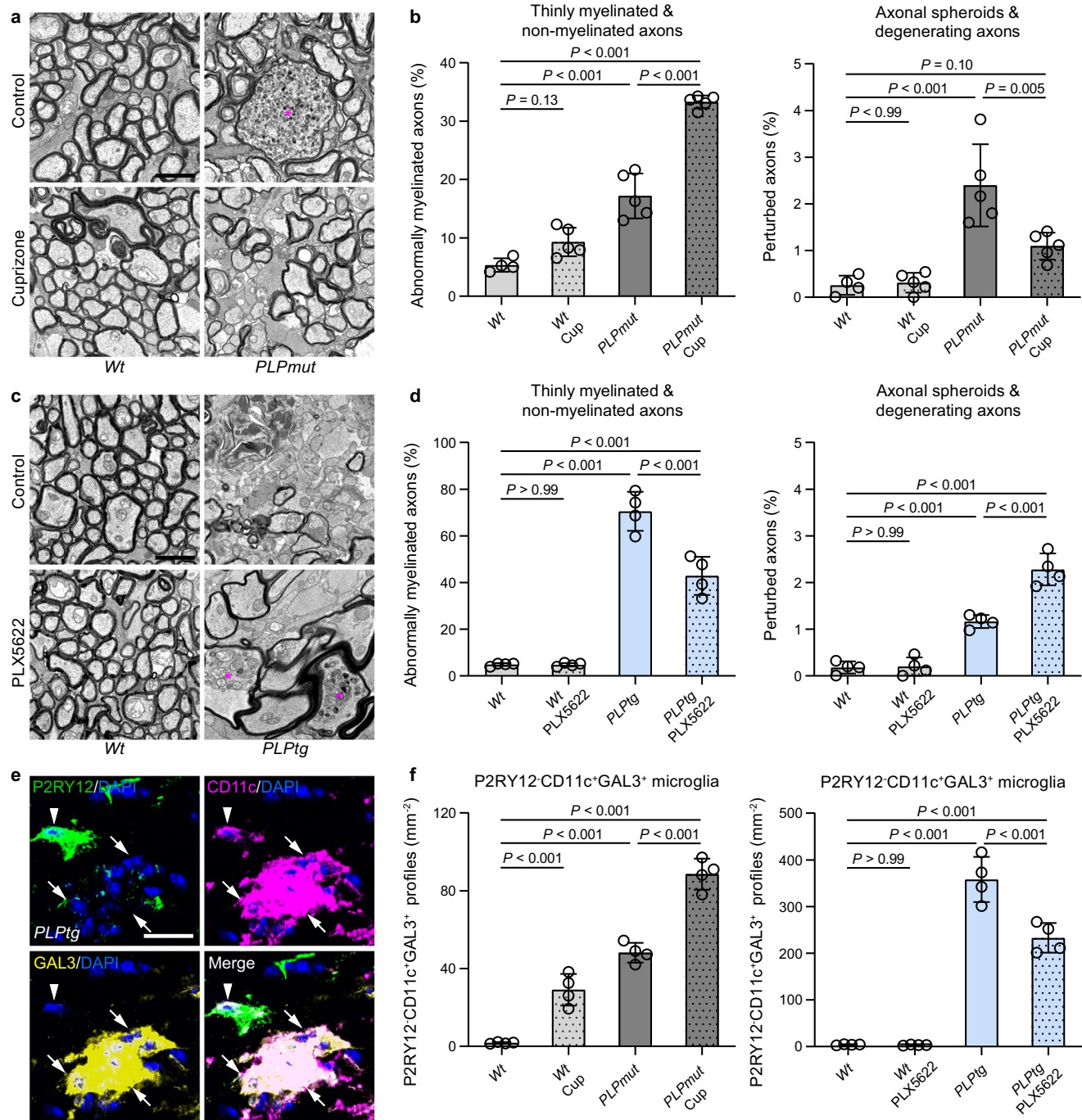

**Fig. 5 | Modulating microglia-mediated removal of perturbed myelin inversely affects axonal damage. a** Representative electron micrographs of optic nerve cross-sections from control (top) or cuprizone treated (bottom) *Wt* (left) and *PLPmut* (right) mice. The asterisk indicates an axonal spheroid. Scale bar, 2 µm. **b** Electron microscopy-based quantification of thinly myelinated (g-ratio ≥ 0.85) and non-myelinated axons (left) or axonal spheroids and degenerating axons (right) in *Wt* and *PLPmut* mice (*n* = 5 mice per group) after control or cuprizone (Cup) diet (each circle represents the mean value of one mouse, *n* = 4,5,5,5 mice per group, one-way ANOVA with Tukey's multiple comparisons test, Left: *F* (3, 15) = 117.5, *P* < 0.001, Right: *F* (3, 15) = 19.13, *P* < 0.001). **c** Representative electron micrographs of optic nerve cross-sections from control (top) or PLX5622 treated (bottom) *Wt* (left) and *PLPtg* (right) mice. The asterisks indicate axonal spheroids. Scale bar, 2 µm. **d** Electron microscopy-based quantification of thinly myelinated (g-ratio ≥ 0.85) and non-myelinated axons (left) or axonal spheroids and degenerating axons (right) in *Wt* and *PLPtg* mice (*n* = 4 mice per group) after control or PLX5622 diet (*n* = 4 mice per group, one-way ANOVA with Tukey's multiple comparisons test, Left: *F* (3, 12) = 118.8, *P* < 0.001, Right: *F* (3, 12) = 82.59, *P* < 0.001). **e** Representative immunofluorescence detection of P2RY12, CD11c, and GAL3 reactivity of microglia in optic nerves. Arrowheads indicate P2RY12+CD11c+GAL3- microglia (AMG1) and arrows indicate P2RY12-CD11c+GAL3+ microglia (AMG2). Scale bar, 20 µm. **f** Quantification of P2RY12-CD11c+GAL3+ microglia in the optic nerves of *Wt* and *PLPmut* after control or cuprizone (Cup) diet (left) and *Wt* and *PLPtg* mice after control or PLX5622 diet (right). AMG2 accumulate in *PLPmut* mice after cuprizone diet and is reduced in number after PLX5622 diet in *PLPtg* mice (*n* = 4 mice per group, one-way ANOVA with Tukey's multiple comparisons test, Left: *F* (3, 12) = 127.4, *P* < 0.001, Right: *F* (3, 12) = 194.9, *P* < 0.001). Data are presented as the mean ± s.d.

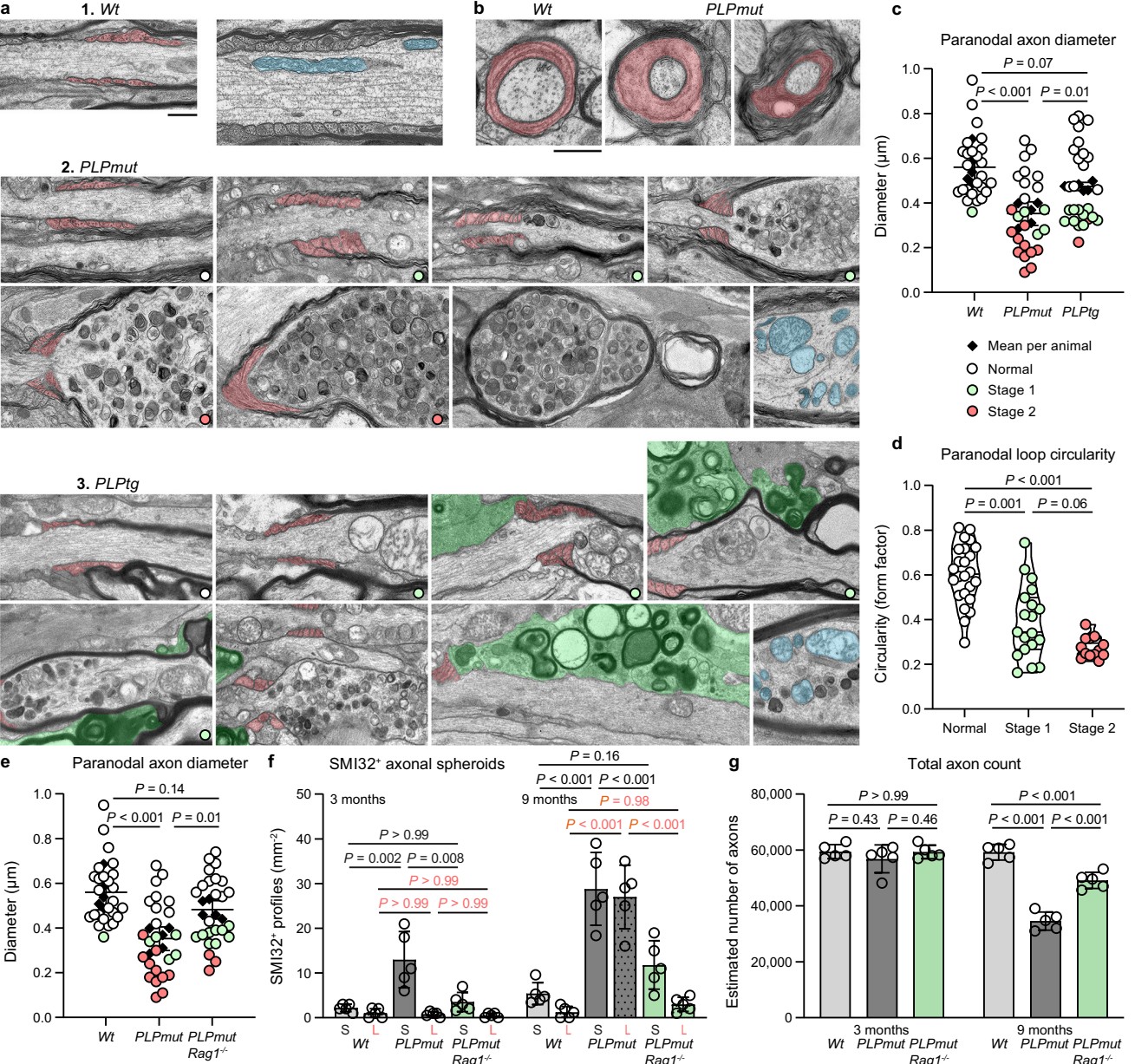

**Fig. 6 | T cell-driven axonal spheroid formation is initiated in proximity to constricted paranodal domains. a** Representative electron micrographs of optic nerve longitudinal sections from 3- to 9-month-old *Wt* (top), *PLPmut* (middle), and *PLPtg* mice displaying putative subsequent stages of axonal spheroid formation. Progressive constriction of axon diameters by oligodendrocytic paranodal loops (light red pseudocolor) correlates with the accumulation of disintegrating mitochondria (exemplarily shown in light blue pseudocolor), juxtaparanodal swelling, and fragmentation in *PLPmut* mice. Early-stage axonal spheroids in *PLPtg* mice are often associated with myelin-phagocytosing microglia (green pseudocolor). Scale bar, 0.5 μm. **b** Fibers with constricted paranodal domains are also detectable in cross-sections of optic nerves from *PLPmut* mice. Scale bar, 0.5 μm **c** Electron microscopy-based quantification of minimum paranodal axon diameters in 9-month-old *Wt*, *PLPmut*, and *PLPtg* mice (each circle represents the value of one paranodal region and each diamond represents the mean value of one mouse), colours indicate stages of spheroid formation, *n* = 5 mice and 25 paranodes per group, one-way ANOVA with Tukey's multiple comparisons tests, $F_{(2, 12)}$ = 17.5,

*P* < 0.001. **d** Analysis of the form factor of paranodal loops reveals decreased circularity along with progressive stages of spheroid formation (each circle represents the mean value of one paranodal region, *n* = 5 mice and 56 paranodes: 24 normal, 19 stage 1, 13 stage 2, one-way ANOVA with Tukey's multiple comparisons test, $F_{(2, 53)}$ = 30.54, *P* < 0.001). **e** Electron microscopy-based quantification of minimum paranodal axon diameters in 9-month-old *Wt*, *PLPmut* (as shown in panel **c**), and *PLPmut/Rag1*[−/−] mice (*n* = 5 mice and 25 paranodes per group, one-way ANOVA with Tukey's multiple comparisons test, $F_{(2, 12)}$ = 15.55, *P* < 0.001). **f** Quantification of small (S) and large (L) SMI32[+] axonal spheroids in 3- and 9-month-old *Wt*, *PLPmut*, and *PLPmut/Rag1*[−/−] mice (each circle represents the mean value of one mouse, *n* = 5 mice per group, two-way ANOVA with Tukey's multiple comparisons test, $F_{(5, 48)}$ = 36.19, *P* < 0.001). **g** Electron microscopy-based estimation of total axonal numbers in the optic nerves. Axon loss is significantly milder in *PLPmut/Rag1*[−/−] than *PLPmut* mice (*n* = 5 mice per group, two-way ANOVA with Tukey's multiple comparisons test, $F_{(2, 24)}$ = 44.33, *P* < 0.001). Data are presented as the mean ± s.d.

axon diameters by measuring CASPR[+] domains using immunofluorescence (Supplementary Fig. 8c). Moreover, in line with previous observations[29,32], absence of functional lymphocytes attenuated the age-dependent formation of small and large axonal spheroids and

axon loss in *PLPmut* mice (Fig. 6f, g). Thus, myelinated fibres under adaptive immune attack seem to constrict their diameters at paranodal domains and thereby hamper axonal transport which can promote axon degeneration[59].

## Cytotoxic effector molecules induce contractile actomyosin activity in myelinating oligodendrocytes

After peripheral nerve injury, Schwann cells form constricting actomyosin spheres to accelerate axon fragmentation in a PAK1-dependent and cytokinesis-like manner[60–62]. Moreover, oligodendrocytes show increased cytoskeletal dynamics after injury and some of the molecules important for myelinating process extension and ensheathment of axons are direct targets for cleavage by the T cell effector protease granzyme B (GZMB)[63–65]. We have previously demonstrated that immune-mediated axonal damage in *PLPmut* and *PLPtg* mice depends on GZMB[30,32] and therefore hypothesized that the focal cytotoxic attack on myelinating oligodendrocyte processes could induce cytoskeletal alterations within non-compacted myelin domains. Supporting this hypothesis, our scRNA-seq analysis of mature ODC from myelin mutants indicated increased expression of several factors related to membrane trafficking and plasma membrane repair (e.g., *Chmp2a*, *Vps4a*, *Syt11*, known to counteract T cell cytotoxicity[66]), regulation of actomyosin structure organization (e.g., *Rhoa*, *Cdc42*, *Pfn1*), and actin filament based process (e.g., *Pak1*, *Wasf2*, *Tpm3*) particularly in *PLPmut* mice (Supplementary Fig. 9a). Labelling of F-actin with phalloidin demonstrated increased levels at the outer aspects of myelin segments around cleaved caspase 3[+] axonal spheroids in optic nerves of *PLPmut* mice, suggesting pro-degenerative signalling in axons likely constricted by oligodendrocytes (Supplementary Fig. 9b, c). Moreover, immunohistochemistry indicated increased diphosphorylated non-muscle myosin light chain (pMLC) levels in cell processes ensheathing SMI32[+] axonal spheroids (Supplementary Fig. 9d), arguing for localized contractile myosin II activity within non-compact myelin domains. Like decreased paranodal diameter, increased pMLC levels in optic nerves of *PLPmut* mice were largely dependent on functional adaptive immunity (Supplementary Figs. 9e, 7a).

Since pMLC has been implicated in the maintenance of the membrane periodic skeleton of axons in vitro[67], we localized the pMLC signal in more detail. Corroborating previous work[68], pMLC was mostly confined to the nodal region flanked by CASPR[+] paranodal domains in optic nerves of *Wt* mice (Fig. 7a). In contrast, CASPR[+] paranodes in *PLPmut* but not *PLPmut/Rag1[−/−]* mice were frequently surrounded by additional spots of pMLC reactivity, especially when showing focal signs of constriction (Fig. 7a, b). To confirm this, we used multiplexed super-resolution fluorescence microscopy by combining 4x physical expansion[69] of optic nerve sections with confocal microscopy and an Airyscan 2 detector. This revealed focal narrowing of the paranodal domains in *PLPmut* mice which were frequently closely enwrapped by compact assemblies of F-actin and pMLC (Fig. 7b, c). Quantification of pMLC[+] assemblies in close association (≤1 μm distance) with individual CASPR[+] paranodes confirmed an immune-dependent accumulation in *PLPmut* mice (Fig. 7c). Since CASPR[+] axonal domains are in direct contact with glial paranodal loops, this suggests local contractile actomyosin activity in non-compact domains of myelinating oligodendrocytes.

To further investigate the relationship between cytotoxic molecules and cytoskeletal dynamics we isolated OPCs from P7 mouse brains and differentiated them for seven days into myelinating oligodendrocytes in vitro. We then stimulated them with a combination of PRF1 and GZMB at sublytic concentrations[70] to mimic a cytotoxic T cell attack (Supplementary Fig. 10a). Six hours after stimulation, a dose-dependent increase in F-actin fluorescence was detected (Supplementary Fig. 10b, e). Live imaging of CellLight Actin-RFP-transfected myelinating ODC confirmed that individual cells showed a relatively stable increase in F-actin levels after stimulation with PRF1 and GZMB (Supplementary Fig. 10c). In addition, pMLC levels increased dose-dependently after stimulation with the cytotoxic molecules (Supplementary Fig. 10d, e). In line with an increased contractile tension of oligodendrocytes after stimulation, MBP[+] cell surface area was decreased. Strikingly, GZMB + PRF1-dependent F-actin and pMLC

induction as well as MBP[+] surface contraction could be inhibited by the simultaneous application of the ROCK inhibitor fasudil (Supplementary Fig. 10f). Our data provide evidence that the immune-driven paranodal constriction of axonal diameters could be mediated by remodelling of the cytoskeleton and actomyosin contractility in myelinating processes.

## Axonal spheroid formation reflects a focal impairment of axonal transport and depends on cytoskeletal dynamics

We and others have previously concluded that axonal spheroids in mice with myelin perturbation form because of focal impairment of axonal transport[29,71]. To confirm this and identify a putative explanation for pro-degenerative signaling in constricted axons we cross-bred *PLPmut* mice with *Nmnat2*-Venus mice which allows live imaging axonal transport of the survival factor in nerve explants[72]. In 12-month-old mice, the total counts of transported NMNAT2 particles as well as their average and maximum velocities were significantly decreased in optic nerve explants of the myelin mutants (Fig. 8a). In contrast, axons in femoral quadriceps nerve explants of *PLPmut* mice showed similar axonal transport efficacy as in *Wt* mice (Fig. 8b), reflecting the CNS-specific impact of PLP defects in mice. Occasionally, we detected focal accumulations of NMNAT2 particles within individual fibres showing little or no ongoing transport in optic nerves of *PLPmut* mice (Fig. 8c). During the 1 h live imaging sessions, these accumulations appeared to increase in frequency, indicating the ongoing formation of obstructions within damaged fibres in explants. Immunofluorescence identified these accumulations of NMNAT2 to occur within SMI32[+] axonal spheroids. When analyzing optic nerves from the same mice before and after the imaging, we confirmed that the frequency of SMI32[+] axonal spheroids was significantly increased after incubating the explants with control medium (Neurobasal A with or without DMSO only) (Fig. 8d). In contrast, when incubating the explants with medium containing cytochalasin D or blebbistatin (dissolved in DMSO) to inhibit actin polymerization or myosin activity, respectively, this increase was prevented.

We next investigated the consequences of these inhibition experiments on pMLC levels in optic nerve explants. Both cytochalasin D and blebbistatin prevented the increased activation of MLC associated with ongoing axonal spheroid formation in *PLPmut* explants (Fig. 9a). Explants from *Wt* mice did not exhibit a significant increase in spheroid formation during the same imaging timeframe (Fig. 9b), confirming that the ongoing immune-driven constriction process depends on perturbed myelinating glia. As a reciprocal experiment we treated explants with calyculin A, a potent protein phosphatase 1 inhibitor that increases myosin contractility. Moreover, we tested the effect of fasudil on optic nerve explants from *PLPmut* mice. While calyculin A further enhanced pMLC levels and axonal spheroid formation, fasudil inhibited both compared with controls (Fig. 9c, d). Finally, we treated *PLPmut* mice for 4 weeks with fasudil in the drinking water to test its efficacy to interfere with ongoing axonal spheroid formation in vivo. Importantly, fasudil did not affect the accumulation of CD8[+] T cells in the white matter of *PLPmut* mice but significantly attenuated the number of axonal spheroids (Fig. 9e), most likely by blocking T cell-driven actomyosin constriction at paranodal domains.

In summary, axonal spheroid formation is associated with a focal impairment of fast axonal transport of NMNAT2, which may explain the activation of pro-degenerative signaling typically observed in programmed axon degeneration[73,74]. Moreover, contractile actomyosin activity (likely occurring in myelinating oligodendrocyte processes) is required for the efficient progression of axonal spheroid formation.

## Discussion

Rare genetic disease models display commonalities with more frequent CNS pathologies and offer unique opportunities to investigate

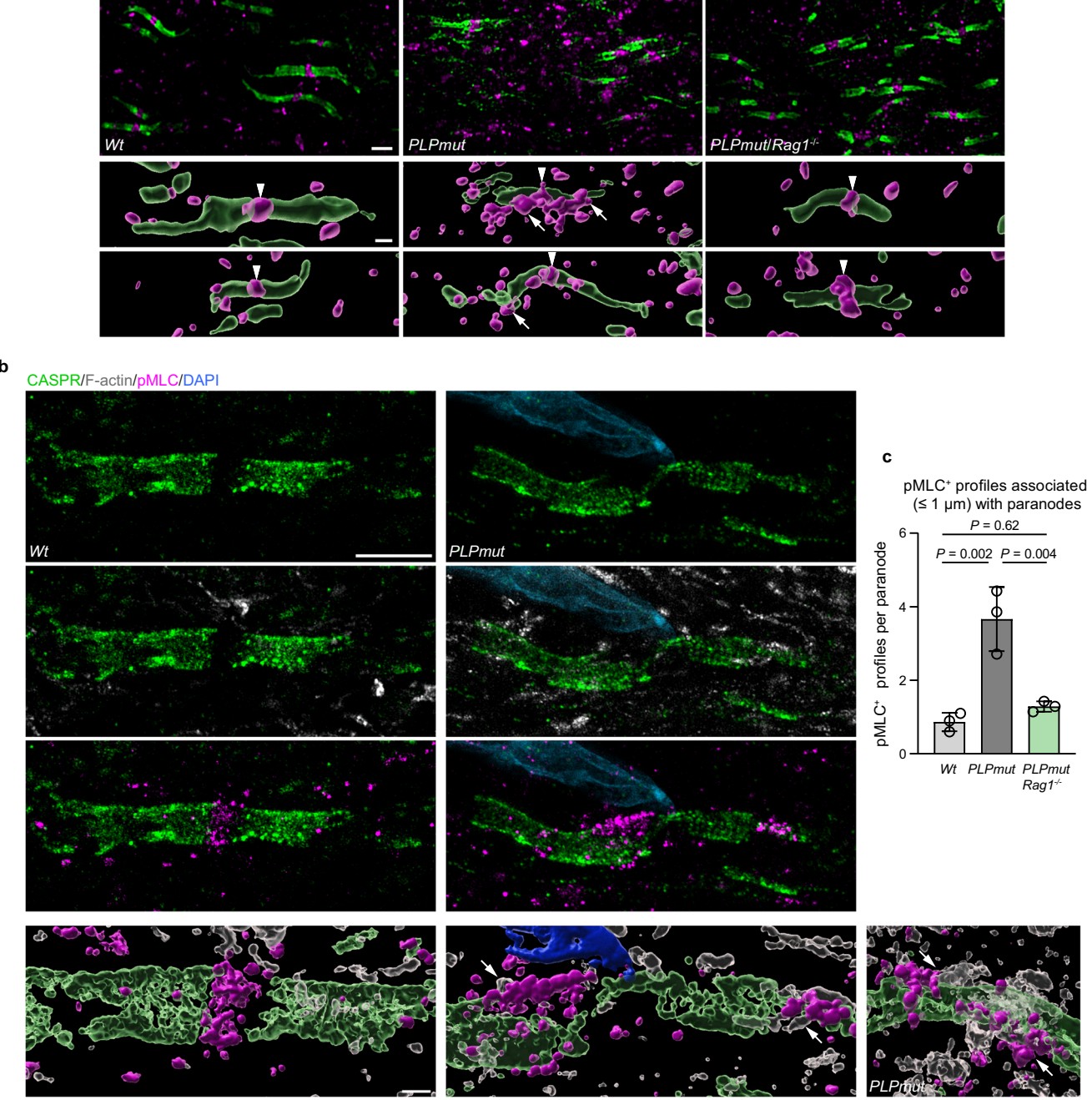

**Fig. 7 | Constricted paranodal domains in myelin mutants are associated with regions showing increased actomyosin activity. a** Representative immuno-fluorescence detection and IMARIS Z-stack surface rendering of CASPR and pMLC in the optic nerves of 12-month-old *Wt*, *PLPmut*, and *PLPmut/Rag1$^{-/-}$* mice. pMLC is localized at normal appearing nodes of Ranvier (arrowheads). In *PLPmut* mice, paranodal domains with irregularly small diameters are surrounded by additional pMLC aggregates (arrows). Scale bar, 2 μm (top), 1 μm (bottom). **b** Super-resolution fluorescence detection and IMARIS Z-stack surface rendering of CASPR, F-actin, and pMLC in the optic nerves of 12-month-old *Wt* and *PLPmut* mice. Compact assemblies of F-actin and pMLC enwrap constricted paranodal domains in *PLPmut*

mice (arrows). Note the proximity of a cell nucleus to the constricted paranodal domain. Scale bars: 10 μm expanded, 2.5 μm unexpanded (top), 2 μm expanded, 500 nm unexpanded (bottom). **c** Quantification of pMLC$^+$ profiles in close asso-ciation (≤1 μm distance) with individual CASPR$^+$ paranodes in optic nerves of *Wt*, *PLPmut*, and *PLPmut/Rag1$^{-/-}$* mice (each circle represents the mean value of one mouse). Paranodes in *PLPmut* mice are associated with a higher number of pMLC$^+$ assemblies which is attenuated by *Rag1* deficiency (n = 25 paranodes per mouse and 3 mice per group, One-way ANOVA with Tukey's multiple comparisons test, $F_{(2, 6)} = 24.27$, P = 0.001). Data are presented as the mean ± s.d.

the consequences of myelin defects in a defined setting of known etiology and progression. Here we studied such models to identify how axonal damage, myelin loss, and neurodegeneration are related to each other and to disability in the context of detrimental cytotoxic T cell reactions. In both *PLPmut* and *PLPtg* mice, previous studies

revealed that CD8$^+$ T cells accumulate in the white matter near nodal domains of myelinated axons[29,31]. While the exact antigen(s) recog-nized by these cells are not identified, they drive myelin degradation and axonal damage in a cognate TCR-dependent manner[30,32]. More-over, a reduction of fast axonal transport efficacy and focal formation

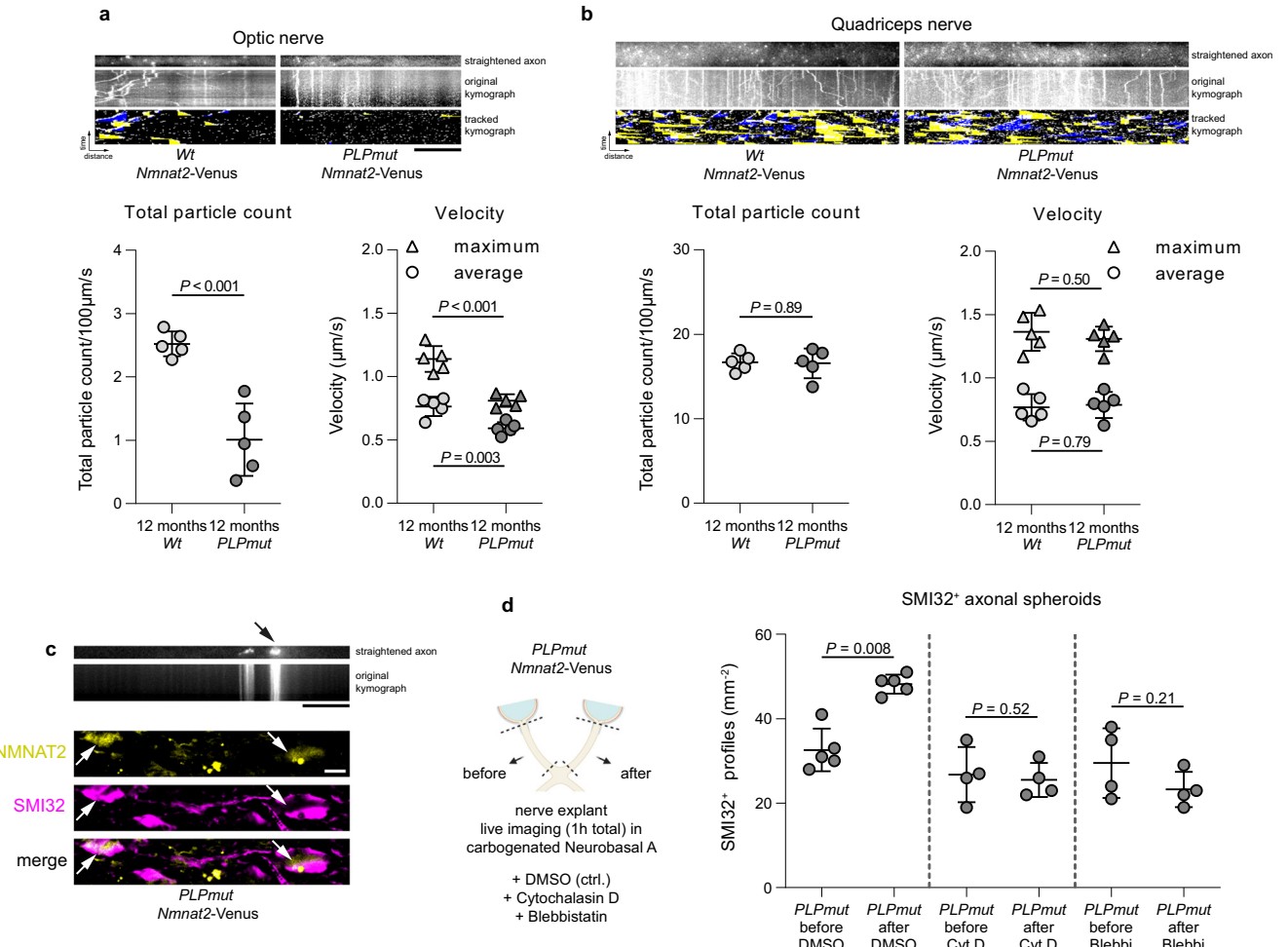

**Fig. 8 | Focal impairment of fast axonal transport in axonal spheroids depends on cytoskeletal plasticity. a** Representative straightened axon, kymograph, and kymograph of successfully tracked particles (top) as well as quantification of axonal transport parameters (bottom) in the optic nerve and **b** femoral quadriceps nerve explants from 12-month-old *Wt/Nmnat2*-Venus and *PLPmut/Nmnat2*-Venus mice. The straightened axon shows the first frame of the time-lapse recording (optic nerve: 60 frames in total, quadriceps nerve: 120 frames in total, frame rate 2 fps). Scale bar, 10 μm. Total NMNAT2 particle count and velocity are decreased in CNS but not PNS axons of *PLPmut* mice (each circle represents the mean value of 5 axons of one mouse, $n = 5$ mice per group, two-sided Student's *t*-test, Optic nerve: Total particle count $t = 5.602$, Maximum velocity $t = 4.241$, Average velocity $t = 6.478$, Quadriceps nerve: Total particle count $t = 0.1369$, Maximum velocity $t = 0.2749$, Average velocity $t = 0.7069$, All d.f. = 8). **c** Straightened axon and kymograph of an optic nerve explant from a *PLPmut/Nmnat2*-Venus mouse exemplifying focal

accumulation of particles (arrow) in an axon with blocked axonal transport (top). Representative immunofluorescence detection of SMI32 reactivity in optic nerves from 12-month-old *PLPmut/Nmnat2*-Venus mice (bottom). The arrows indicate axonal spheroids with an accumulation of NMNAT2 particles. Scale bars, 10 μm. **d** Schematic experimental design (left, created with BioRender.com). SMI32⁺ axonal spheroids were quantified in pairs of optic nerves from *PLPmut/Nmnat2*-Venus mice before or after live imaging explants in neurobasal A medium containing DMSO, cytochalasin D (Cyt D), or blebbistatin (Blebbi). Inhibition of actin polymerization or myosin activity blocks the increased frequency of SMI32⁺ axonal spheroids after 1 h imaging ex vivo (each circle represents the mean value of one optic nerve, $n = 5,4,4$ mice per group, two-sided paired *t* test, DMSO: $t = 4.968$, d.f. = 4, Cyt D: $t = 0.7346$, d.f. = 3, Blebbi: $t = 1.575$, d.f. = 3). Data are presented as the mean ± s.d.

of juxtaparanodal axonal spheroids are shared features at early stages of immune-mediated damage.

In our current study we observed that these spheroids initially form in axonal segments ensheathed by perturbed myelin. Interestingly, axonal spheroids only progress in size and show further signs of axon degeneration when the myelin sheath remains attached. Accordingly, neurodegeneration and clinical disease are inversely related to the degree of demyelination in the distinct disease models. Thus, efficient removal of perturbed myelin segments may allow resilience or recovery of axons at early stages of damage and argues against the current dogma that axon degeneration in myelin disease is mostly secondary to loss of myelin. Nevertheless, there is evidence that the long-term failure to regenerate myelin correlates strongly with neurodegeneration and disability in MS[75–77]. Since previous observations in EAE have shown that focal axonal damage occurs in myelinated

fibres and can be reversible[57], it seems important that perturbed myelin is rapidly removed and replaced by new myelin sheaths. A recent study in human MS and experimental models of inflammatory demyelination is consistent with the notion that myelin itself poses a risk for axon degeneration[78]. Oligodendrocytes under immune attack were proposed to provide insufficient metabolic support to underlying axons and thereby promote their degeneration, which might be reversed by rapid demyelination. Here we describe an additional, mechanical process of how oligodendrocytes targeted by CD8⁺ T cells gain properties detrimental to axonal integrity that could explain focal damage and impairment of axonal transport near the nodal domains.

Myelinating glia in the peripheral nervous system constrict axons after injury to facilitate fragmentation, degradation, and subsequent regeneration[60–62]. This process depends on cytoskeletal dynamics comprising actin-myosin remodelling and is instructed by altered

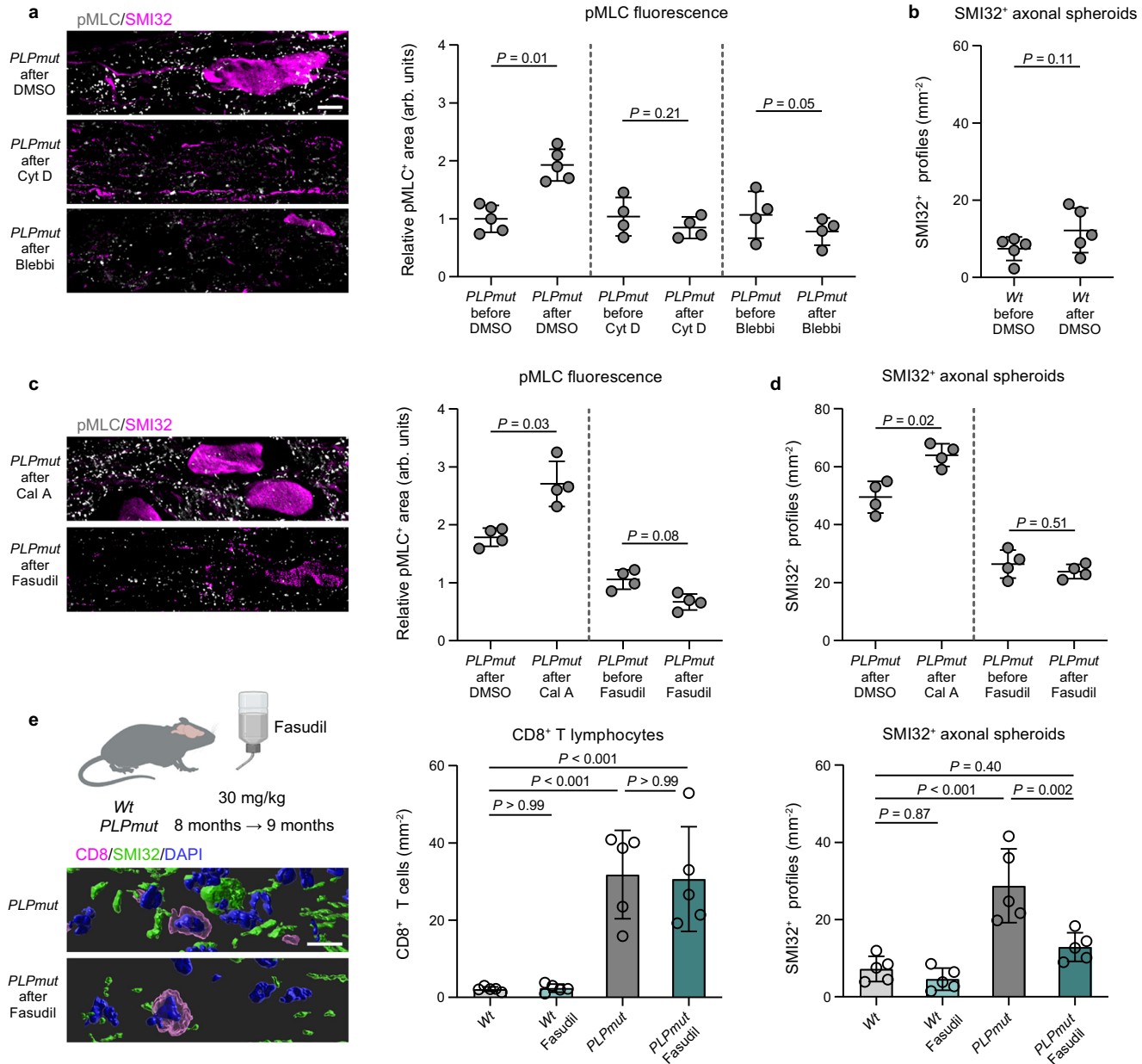

**Fig. 9 | Actomyosin constriction drives axonal spheroid formation in myelin mutant mice. a** Representative immunofluorescence detection (left) of pMLC and SMI32 in optic nerve explants from 12-month-old *PLPmut/Nmnat2*-Venus mice after live imaging explants in neurobasal A medium containing DMSO, cytochalasin D (Cyt D), or blebbistatin (Blebbi). Quantification (right) of pMLC fluorescence by thresholding analysis demonstrates an increase in relative pMLC$^+$ area after live imaging which is blocked by Cyt D or Blebbi (each circle represents the mean value of one optic nerve, *n* = 5,4,4 mice per group, two-sided paired *t* test, DMSO: *t* = 4.292, d.f. = 4, Cyt D: *t* = 1.613, d.f. = 3, Blebbi: *t* = 3.125, d.f. = 3). **b** Quantification of SMI32$^+$ axonal spheroids in pairs of optic nerves from *Wt/Nmnat2*-Venus mice before and after live imaging explants for 1 h in neurobasal A medium containing DMSO (each circle represents the mean value of one optic nerve, *n* = 5 mice per group, two-sided paired *t* test, *t* = 2.053, d.f. = 4). **c** Representative immuno-fluorescence detection (left) of pMLC and SMI32 in optic nerve explants from 12-month-old *PLPmut* mice after maintaining explants in neurobasal A medium

containing DMSO, calyculin A (Cal A), or fasudil. Quantification (right) of pMLC fluorescence by thresholding analysis and **d** SMI32$^+$ axonal spheroids demonstrates a further increase of relative pMLC$^+$ area and axonal spheroid formation after Cal A and a block of pMLC induction and axonal spheroid formation after fasudil (each circle represents the mean value of one optic nerve, *n* = 4 mice per group, two-sided paired *t* test, Cal A: *t* = 3.703, d.f. = 3, fasudil: *t* = 2.577, d.f. = 3) **e** Schematic experimental design (left, created with BioRender.com) and IMARIS Z-stack surface rendering of CD8 and SMI32 in the optic nerves of 9-month-old *PLPmut* mice with or without fasudil treatment. Scale bar, 10 μm. Quantification of CD8$^+$ T cells (middle) and SMI32$^+$ axonal spheroids (right) demonstrates no impact of fasudil on the numbers of CD8$^+$ T cells but on axonal spheroid formation in *PLPmut* mice (each circle represents the mean value of one optic nerve, *n* = 5 mice per group, One-way ANOVA with Tukey's multiple comparisons test, CD8: *F* (3, 16) = 17.72, *P* < 0.001, SMI32: *F* (3, 16) = 18.99, *P* < 0.001). Data are presented as the mean ± s.d.

interactions between severed axons and Schwann cells. Recent findings suggest that oligodendrocytes with lowered Dusp6 expression can fulfil similar functions and enable repair after injury[79], indicating some capacity for dedifferentiation[80]. Oligodendroglial Vangl2, a RhoA-Myosin II-dependent planar polarity protein, controls paranodal

axon diameter[81], demonstrating compressive action of the paranodal spiral on the axon. Developmental process extension by oligodendrocytes at the leading edge of myelin sheaths involves actin dynamics[64,82]. Oligodendrocytes also increase cytoskeletal plasticity after inflammatory damage, which is associated with early changes at

the paranodal domains[63,83]. Our data indicate that increased acto-myosin contractility at the paranodal domains can be detrimental to axonal integrity in non-lesioned conditions, which is in line with the vulnerability of fast axonal transport to mechanical constriction[84] and could explain the high vulnerability of small diameter myelinated axons. Using various inhibitors to decrease or enhance actomyosin activity in an ex vivo explant system and in vivo, we demonstrate its involvement in axonal spheroid formation. We do not know where exactly the contractile forces are generated, but there are several findings that point to the paranodal glial domains: First, super-resolution microscopy showed that compact assemblies of F-actin and pMLC form directly adjacent to constricted paranodal axon domains in *PLPmut* mice. Second, a direct mode of action on the neuron is difficult to explain since the membrane periodic skeleton maintains axonal integrity and its disruption via cytoskeletal inhibitors accelerates beading and fragmentation in vitro[85,86]. Third, our experiments using *Wt* explants, as well as the spatiotemporal dynamics of spheroid for-mation and the above-described similarities with myelinating glia in the peripheral nervous system suggest an oligodendrocyte-related mechanism. The predominant localization of cytoskeletal elements within non-compact myelin domains is consistent with the observed paranodal narrowing or "strangulation" of the axonal partners and the accumulation and disintegration of organelles at juxtaparanodal domains[64]. Nevertheless, we can presently not fully exclude that actomyosin activity within the axon is involved in the constriction.

We observed that the combined cytotoxic effector molecules perforin and granzyme B induce the cytoskeletal alterations and the reduction of the myelin membrane sheet surface area in oligoden-drocyte cultures. This was blocked by fasudil, an established inhibitor of ROCK-dependent contractility. Moreover, increased pMLC levels in the white matter, paranodal constriction, and axonal spheroid for-mation in *PLPmut* mice were attenuated by immune deficiency. Since *PLPmut/Rag1$^{-/-}$* mice still exhibit structurally altered myelin, we con-clude that these changes are not a cell-intrinsic consequence of the myelin perturbation. Granzyme B can directly cleave kinases and binding proteins important for controlling cytoskeletal dynamics and thereby induce membrane blebbing in target cells[65]. Such plasma-lemmal protrusions are also important for physiological processes like cytokinesis and locomotion and can provide resilience against cell lysis and necrosis[87]. It is therefore possible that CD8$^+$ T cells directly target myelin segments of perturbed oligodendrocytes and thereby induce cytoskeletal alterations, eventually resulting in abnormal constriction of axons at their paranodal domains. This mechanism could provide an explanation for why persistent ensheathment with mutant myelin ultimately results in axonal transection, whereas demyelination abol-ishes the immune target(s) as well as the constricting oligodendroglial process.

Why oligodendrocytes in the investigated mouse models show distinct reactions upon the CD8$^+$ T cell attack remains an open ques-tion. While increased expression of disease-associated pro-inflamma-tory transcripts and of ESCRT-mediated membrane repair components is found in both models, upregulation of molecules protecting against phagocytosis and cell death is specific to *PLPmut* mice. This is associated with higher levels of actomyosin-related cytoskeletal molecules and axon degeneration. In contrast, there is increased expression of "eat me" signals in *PLPtg* oligodendrocytes, resulting in the formation of a phagocytic, axon-protective microglia state (AMG2). Similar to the PNS[88], the CSF-1-CSF-1R axis seems to play an important part in the communication between neural cells and white matter-resident phagocytes. By modulating microglia-mediated myelin removal, we show that these reactions can be fostered or attenuated, indicating that they are not exclusively driven by oligodendrocyte-intrinsic differences caused by distinct gene defects. Since both axon degeneration and demyelination are strongly mod-ified by CD8$^+$ T cells, these differential glial reactions may instead be

induced by distinct subpopulations of CD8$^+$ T cells that utilize different effector molecules. In line with this hypothesis, a recent study has shown that aging-related transcriptional responses in oligoden-drocytes are at least partially dependent on the presence of CD8$^+$ T cells in the white matter[8]. Mild oligodendrocyte cell loss and demyelination were related to an interferon response, while a distinct state of aged oligodendrocytes was characterized by increased expression of *Serpina3n*, an inhibitor of granzyme B. We therefore hypothesize that oligodendrocytes try to resist granzyme B and con-strict axons but instruct microglia-mediated demyelination upon interferon challenge. This hypothesis is in line with the recently described heterogeneity of CNS-associated CD8$^+$ T cells and their effector molecules in myelin mutants and aged mice and is supported by the observation that *Gzmb* deficiency in adaptive immune cells attenuates axonal perturbation but not demyelination in *PLPtg* mice[7,30,32]. Moreover, we observed that several transcripts specific for the interferon-responsive oligodendrocyte state (e.g., *Ifi27l2a*, *B2m*) are upregulated in ODC from *PLPtg* but not *PLPmut* mice.

In *PLPtg* mice, microglia associated with demyelination (AMG2) showed gene set enrichment for oxidative phosphorylation, typical for tolerogenic antigen presenting cells and phagocytes[89,90]. Moreover, they upregulated marker genes indicating axon protection and downregulated genes important for T cell stimulation. Thus, they seem to play a beneficial role in removing perturbed myelin and promoting the resolution of inflammation in demyelinated regions which is in line with the preferential accumulation of CD8$^+$ T cells in still myelinated regions. However, these microglia also downregulate genes important for lipid export and show accumulation of lysosomal storage material. Thus, overt myelin degradation seems to burden these microglia and might explain the limited remyelination capacity in *PLPtg* mice[91]. On the other hand, a disease-associated microglia state (AMG1) with pro-inflammatory properties was detected in both myelin disease models, reflecting the context-specific reaction of microglia, and explaining the different outcome after pharmacological microglia depletion (this study and[34]). Targeting microglia can thus be both beneficial by attenuating neuroinflammatory processes (as observed in *PLPmut* mice) and detrimental by promoting dysfunction of protective reac-tions (as observed in *PLPtg* mice).

Our observations reveal that focal impairment of axonal transport and formation of axonal spheroids at juxtaparanodal domains is induced by T cell-driven paranodal alterations. Interestingly, this is associated with the activation of pro-degenerative signalling (cleaved caspase 3) in axons, reminiscent of trophic deprivation[74]. Since these reactions also occur to some extent in *PLPtg* mice but are less evident in demyelinated regions, they do not necessarily result in axon loss. However, mitochondrial defects and decreased axonal transport of the survival factor NMNAT2 are known to facilitate Wallerian-like axon degeneration[73]. This indicates that later stages of neuroinflammation-driven axonal damage in *PLPmut* mice are likely to be executed by SARM1-dependent, active signalling within axons. While a previous study did not find protection of axons in *Plp1* knockout mice upon crossbreeding with the *Wld$^S$* line[71], impaired axonal transport could have prevented axonal delivery of the fusion protein. Since we some-times observed consecutive spheroids along the same axons, we pro-pose that chronic focal damage at multiple myelinated target sites could add up to promote axon degeneration and subsequent neuron loss. Future experiments should aim to clarify the molecular mechanisms of neurodegeneration in *PLPmut* mice.

In conclusion, our data show that while myelination offers many advantages for higher functions of the nervous system, it also poses a risk for neuroinflammation-related axonal damage and degeneration when oligodendrocytes become dysfunctional. The evolutionary benefit of myelin appears to outweigh this concomitant risk for insu-lated axons in pathological conditions. Clearly, demyelination per se is generally not beneficial but could provide resilience of compromised

fibers allowing survival to possibly enable restoration, as demonstrated in distinct myelin disease models. In this context, a major open question relates to the reason for the susceptibility of myelin perturbation to induce adaptive immune reactions. Oligodendrocytes burdened by myelin defects may incite neuroinflammation as a conserved mechanism to eliminate stressed or dysfunctional cells. Additionally, mutant oligodendrocytes targeted by CD8[+] T cells might constrict axons by recapitulating a glial program beneficial for the clearance of debris and regeneration in injury conditions, similar to Schwann cells during Wallerian degeneration. Our findings are of translational relevance for therapy approaches to modulate neuroinflammation in myelin disease. While they imply that blocking detrimental immune reactions can have benefits in chronic progressive disease, they also show that certain aspects of neuroinflammation preserve axonal integrity under compromised conditions. Greater insights are needed to identify strategies to block detrimental but still allow or even foster beneficial neural-immune interactions to confer resilience and possibly enable recovery of the perturbed white matter.

## Methods

### Animals

All animal experiments were approved by the Government of Lower Franconia, Würzburg, Germany (AZ 55.2 DMS 2532-2-399; AZ 55.2 DMS 2532-2-399; AZ 55.2 DMS 2532-2-907; AZ 55.2 DMS 2532-2-1029; AZ 55.2 DMS 2532-2-1347). Mice were kept at the animal facility of the Centre for Experimental Molecular Medicine, University of Würzburg, under barrier conditions and at a constant cycle of 14 h in the light ( < 300 lux) and 10 h in the dark. Colonies were maintained at 20−24 °C and 40−60% humidity, with free access to food and water. All mice including wild type (*Wt*), homozygous *PLPmut* (R137W, B6.Cg-Tg(PLP1)1Rm-*Plp1*^tm1Kan^/J)[29] - genuine or crossbred with *Rag1*^−/−^ (B6.129S7-*Rag1*^tm1Mom^/J)[92], hemizygous *PLPtg* (B6.Cg-Tg(Plp1)66Kan/J)[37], mice were on a uniform C57BL/6 J genetic background; they were bred, regularly backcrossed and aged in-house. *Nmnat2*-Venus[93] mice were cross-bred with *PLPmut* mice. *Wt* and *PLPmut* offspring hemizygous for the *Nmnat2* transgene were analyzed. Since we did not detect obvious differences between male and female mice in the analyses presented in the current study, mice of either sex were used for the experiments. Genotypes were determined by conventional PCR using isolated DNA from ear punch biopsies.

### Accelerating rotarod analysis

Mice were placed on a RotaRod Advanced system (TSE systems); the time on the constantly accelerating rod (5−50 r.p.m., max latency 300 s) was measured in 5 consecutive runs per trial as described previously[29]. Mice were trained with two trials on two consecutive days and latencies were measured in a third trial on the third day.

### Analysis of visual acuity

The visual acuity of mice was analyzed using automated optokinetic reflex tracking in an OptoDrum device (Striatech). Briefly, mice were placed on an elevated platform surrounded by monitors and a stripe pattern with maximum contrast and constant rotation speed (12 deg s$^{-1}$) was presented. Behavior was automatically detected and analyzed by the OptoDrum software v.1.2.6 in an unbiased manner and the stimulus pattern (cycles) was continuously adjusted to find the threshold of the animal's visual acuity.

### Spectral domain optical coherence tomography (OCT)

Mice were subjected to OCT imaging with a commercially available device (SPECTRALIS OCT; Heidelberg Engineering) and additional lenses as described previously[29]. Mice were measured at different ages for longitudinal analysis and the thickness of the innermost retinal composite layer comprising the nerve fibre layer (NFL), GCL and inner plexiform layer (IPL) were measured in high-resolution peripapillary circle scans (at least ten measurements per scan) by an investigator unaware of the genotype and treatment condition of the mice using HEYEX v.1.7.1.

### Histochemistry and immunofluorescence

Mice were euthanized with $CO_2$ (according to the guidelines by the State Office of Health and Social Affairs Berlin), blood was removed by transcardial perfusion with PBS containing heparin and tissue was fixed by perfusion with 2% paraformaldehyde (PFA) in PBS. Tissue was collected, postfixed, dehydrated, and processed as described previously[29]. Immunohistochemistry was performed on 10- or 30-µm-thick longitudinal optic nerve and coronal brain sections after post-fixation in 4% PFA in PBS or ice-cold acetone for 10 min. Sections were blocked using 5% BSA in PBS and incubated overnight at 4 °C with an appropriate combination of up to 3 of the following antibodies or stains: mouse anti-neurofilament H non-phosphorylated, SMI32 (1:1,000, catalog no. 801701; BioLegend); rat anti-CD8 (1:500, catalog no. MCA609G; Bio-Rad Laboratories); rabbit anti-MBP (1:300, catalog no. PD004, MBL); rat anti-CD11b (1:100, catalog no. MCA74G; Bio-Rad Laboratories); hamster anti-CD11c (1:100, catalog no. MA11C5; Thermo Fisher Scientific); rabbit anti-P2RY12 (1:300, catalog no. 55043 A, Ana-Spec); rabbit anti-GAL3 (1:1,000, NBP3-03252; Novus Biologicals); rat anti-GAL3 (1:300, catalog no. 125402, BioLegend); phalloidin-TRITC (1:300, catalog no. P1951; Sigma-Aldrich); FluoroMyelin Green (1:300, catalog no. F34651, Thermo Fisher Scientific); rabbit anti-cleaved caspase 3 (1:300, catalog no. 9664; Cell Signaling); rabbit anti-pMLC Thr18/Ser19 (1:100, catalog no. 3674; Cell Signaling); mouse anti-CASPR (1:1000, catalog no. 75-001; NeuroMab); goat anti-IBA1 (1:300, catalog no. NB100-1028; Novus Biologicals); goat anti-ICAM1(1:1,000, catalog no. AF796; Novus Biologicals); rabbit anti-laminin (1:300, catalog no. ab11575; Abcam), rabbit anti-c-Jun (1:300, catalog no. 9165; Cell Signaling); rat anti-MHC-I (1:100, catalog no. T-2105; Dianova); Immunoreactive profiles were visualized using fluorescently labeled (1:300; Dianova) secondary antibodies, streptavidin (1:300; Thermo Fisher Scientific) or biotinylated secondary antibodies (1:100; Vector Laboratories) and streptavidin-biotin-peroxidase (Vector Laboratories) complex using diaminobenzidine HCl and $H_2O_2$; nuclei were stained with 4,6-diamidino-2-phenylindole (DAPI) (Sigma-Aldrich). Light and fluorescence microscopy images were acquired using an Axio Imager M2 microscope (ZEISS) with ApoTome.2 structured illumination equipment, attached Axiocam cameras and corresponding software (ZEN v.2.3 blue edition) or a FluoView FV1000 confocal microscope (Olympus) with corresponding software (v.2.0). Images were minimally processed (rotation, cropping, addition of symbols) to generate figures using Photoshop CS6 and Illustrator CS6 (Adobe). Z-stack surface rendering was performed using IMARIS v.9.7 (Bitplane). For quantification, immunoreactive profiles were counted in at least three nonadjacent sections for each animal and related to the area of these sections using the cell counter plugin in Fiji/ImageJ v.1.51 (National Institutes of Health). To quantify RGCs, perfusion-fixed eyes were enucleated, and specific markers of the inner retinal cell types were labelled in free-floating retina preparations. Fixed retinae were frozen in PBS containing 2% Triton X-100, thawed, washed, and blocked for 1 h using 5% BSA and 5% donkey serum in PBS containing 2% Triton X-100. Retinae were incubated overnight on a rocker at 4 °C with appropriate combinations of the following antibodies: guinea pig anti-RBPMS (1:300, catalog no. ABN1376; Merck Millipore); goat anti-BRN3a (1:100, catalog no. sc-31984; Santa Cruz Biotechnology); immune reactions were visualized using fluorescently labelled (1:500; Dianova) secondary antibodies, retinae were flat-mounted, and the total retinal area was measured. RGCs were quantified in three images of the middle retinal region per flat mount using the cell counter plugin in Fiji/ImageJ v.1.51 (National Institutes of Health).

## Electron microscopy

The optic nerves of transcardially perfused mice were postfixed overnight in 4% PFA and 2% glutaraldehyde in cacodylate buffer. Nerves were osmicated and processed for light and electron microscopy; morphometric quantification of neuropathological alterations was performed as published previously[29] using a LEO906 E electron microscope (ZEISS) and corresponding software iTEM v.5.1 (Soft Imaging System). At least 10 regions of interest (corresponding to an area of around 5% and up to 3000 axons per individual optic nerve) were analyzed per optic nerve per mouse. The percentages of axonal profiles showing spheroid formation or undergoing degeneration were identified individually by their characteristic morphological features in electron micrographs and related to the number of all investigated axons per optic nerve per mouse. Images were processed (rotation, cropping, addition of symbols and pseudocolor) to generate figures using Photoshop CS6 and Illustrator CS6 (Adobe).

## Flow cytometry and cell sorting

Mice were euthanized with $CO_2$ (according to the guidelines by the State Office of Health and Social Affairs Berlin) and blood was thoroughly removed by transcardial perfusion with PBS containing heparin. Brains including optic nerves, leptomeninges and choroid plexus were dissected, collected in ice-cold PBS and cut into small pieces. Tissue was digested in 1 ml of Accutase (Merck Millipore) per brain at room temperature for 15 min and triturated through 70-µm cell strainers, which were rinsed with 10% FCS in PBS. Cells were purified by a linear 40% Percoll (GE Healthcare) centrifugation step at 650 g without brakes for 25 min and the myelin top layer and supernatant were discarded. Mononuclear cells were resuspended in 1% BSA in PBS and isolated cells were counted for each brain. For scRNA-seq, cells from the brains of 3 adult (10-month-old) *Wt*, 3 *PLPmut*, and 3 *PLPtg* mice were analyzed. Viable cells were identified by Calcein blue AM stain (catalog no. ABD-22007; Biomol), Fc receptors were blocked for 15 min with rat anti-CD16/32 (1:200, catalog no. 553141; BD Biosciences) and cells were washed and labelled with the following antibodies for 30 min at 4 °C: rat anti-CD45 PerCP/Cyanine5.5 (1:100, catalog no. 130-102-469; Miltenyi Biotec); rat anti-SiglecH PE (1:100, catalog no. 12-0333-82; eBioscience); rat anti-O1 AF700 (1:100, FAB1327N-100UG; R&D Systems). Cells were washed twice, single viable cells were gated and CD45⁻O1⁺ and CD45^low Siglec-H⁺ cells were collected using a FACSAria III and corresponding software (FACSDiva, v.6; BD Biosciences). Calculation of the number of CD45^low SiglecH⁺ microglia per brain was performed by extrapolating their frequency to the counted total number of isolated cells. For further experiments viable CD45^low Siglec-H⁺ microglia were labelled with rat anti-ICAM1 BV711 (1:100, catalog no. 116143; BioLegend), rat anti-Ly6A/E FITC (1:100, catalog no. 108106; BioLegend), rat anti-CD11c APC (1:100, catalog no. 117310: BioLegend), and rat anti-PD1 BV605 (1:100, catalog no. 135219; BioLegend). Cells were washed twice; single viable cells were gated and CD45^low Siglec-H⁺ cells were analyzed using a FACSLyric (BD Biosciences) and Flowjo (version 10).

## Single-cell RNA sequencing (scRNA-seq) and data processing

Around 15,000 CD45^low Siglec-H⁺ and CD45⁻O1⁺ single cells each was sorted per sample using a FACSAria III (BD Biosciences) before being encapsulated into droplets with the Chromium Controller (10x Genomics) and processed according to the manufacturer's specifications. Briefly, every transcript captured in all the cells encapsulated with a bead was uniquely barcoded using a combination of a 16-base pair (bp) 10x barcode and a 10-bp unique molecular identifier (UMI). Complementary DNA libraries ready for sequencing on Illumina platforms were generated using the Chromium Single Cell 3' Library & Gel Bead Kit v2 (10x Genomics) according to the detailed protocol provided by the manufacturer. Libraries were quantified by Qubit 3.0 Fluorometer (Thermo Fisher Scientific) and quality was checked using a 2100

Bioanalyzer with High Sensitivity DNA kit (Agilent Technologies). Libraries were pooled and sequenced with a NovaSeq 6000 platform (S1 Cartridge; Illumina) in paired-end mode to reach a mean of 44,131 reads per single cell. A total of 13,556, 9,955, and 12,191 cells were captured and a median gene number per cell of 2268, 2562, and 2165 could be retrieved for adult *Wt*, *PLPmut*, and *PLPtg* cells, respectively. Data were demultiplexed using the CellRanger software v.2.0.2 based on 8 bp 10x sample indexes; paired-end FASTQ files were generated. The cell barcodes and transcript unique molecular identifiers were processed as described previously[94]. The reads were aligned to the University of California, Santa Cruz mouse mm10 reference genome using STAR aligner v.2.5.1b. The alignment results were used to quantify the expression level of mouse genes and generate the gene-barcode matrix. Subsequent data analysis was performed using the R package Seurat[95] v.4.0. Doublets and potentially dead cells were removed based on the percentage of mitochondrial genes (cutoff set at 60%) and the number of genes (cells with >100 and <6,000 genes were used) expressed in each cell as quality control markers. The gene expression of the remaining cells (13,522, 9898, and 12,170 cells from *Wt*, *PLPmut*, and *PLPtg* mice, respectively) was log-normalized. Highly variable genes were detected with Seurat and the top 2000 of these genes were used as the basis for downstream clustering analysis. Data were scaled, principle component analysis was used for dimensionality reduction and the number of principal components was identified using the built-in Elbow plot function. Cells were clustered based on the identified principal components (10) with a resolution of 0.5; uniform manifold approximation and projection were used for data visualization in two dimensions. Microglia and oligodendrocyte clusters were subset based on marker gene expression and reanalyzed, resulting in 9484, 8405, and 8419 cells from *Wt*, *PLPmut*, and *PLPtg* mice, respectively. Due to their vulnerability during the processing after sorting, much lower numbers of oligodendrocytes compared with microglia remained in the resulting dataset. These were reanalyzed and reclustered based on 14 principal components and a resolution of 0.6. The contribution of the samples to each microglia cluster in absolute numbers was calculated by extrapolating their frequencies to the number of CD45^low Siglec-H⁺ cells per brain. Differentially expressed genes were identified using the FindMarkers function with min.pct = 0.25. Complete lists of differentially expressed genes ($p\_val\_adj > 0.05$ after Bonferroni or FDR correction) are included in Supplementary Data 1. Marker gene scores for feature expression programs were calculated using the AddModuleScore function in Seurat. Gene-set enrichment analysis was performed using Metascape v.3.5 (http://metascape.org)[96].

## Cuprizone treatment

Demyelination was induced by feeding 10- to 12-week-old *Wt* and *PLPmut* mice a diet containing 0.2% cuprizone (bis-cyclohexanone oxaldihydrazone; Sigma-Aldrich) in ground standard rodent chow for 6 weeks. Control mice were fed standard rodent chow. Optic nerves and corpora callosa were processed for immunofluorescence and electron microscopy. The procedures were approved by the Review Board for the Care of Animal Subjects of the district government (Niedersächsisches Landesamt für Verbraucherschutz und Lebensmittelsicherheit, Oldenburg, Germany, AZ 33.9 42502-04-15/1762) and performed according to international guidelines on the use of laboratory mice.

## PLX5622 treatment

PLX5622 (provided by Plexxikon Inc., Berkeley, CA, USA) was prepared as a 300 ppm drug chow to dose ~54 mg PLX5622/kg body weight when given *ad libitum*. This was based on our previous long-term treatment approaches in which we observed efficient microglia depletion without obvious neurological side effects in *Wt* and *PLPmut* mice[34]. Control mice received normal chow without the pharmacological inhibitor. *Wt* and *PLPtg* mice were treated for 6 months with daily

monitoring concerning certain burden criteria and phenotypic abnormalities. The treatment started at 6 months of age before prominent demyelination in *PLPtg* mice.

## RNAscope

Multiplex RNAscope was performed based on the manufacturer's (Advanced Cell Diagnostics) instructions[97]. Briefly, sections were postfixed in ice-cold 4% PFA for 30 min. Sections were then dehydrated using a series of ethanol solutions (50 - 100%) before drying and boiling for 5 min in fresh target retrieval solution at 100 °C. Slides were washed in distilled water and rinsed with 100% ethanol before incubation with protease III for 30 min at 40 °C. Slides were washed again and hybridized with gene-specific probes against *Csf1* and *Mbp* for 2 h at 40 °C in a HybEZ oven (ACD). Non-annealed probes were removed by washing sections in 1x proprietary wash buffer, and the signal was amplified with 4 amplification systems (Amp1 - Amp4). Finally, sections were stained with DAPI, mounted and analyzed using an Axio Imager M2 microscope (ZEISS) with ApoTome.2 structured illumination equipment, attached Axiocam cameras and corresponding software (ZEN v.2.3 blue edition).

## Expansion microscopy

Super-resolution fluorescence microscopy was performed with 30-μm-thick longitudinal optic nerve cryo-sections. Free-floating sections were blocked using 5% BSA and 5% donkey serum in PBS containing 0.3% Triton X-100 and incubated overnight at 4 °C with a combination of the following antibodies and stains: rabbit anti-pMLC Thr18/Ser19 (1:100, catalog no. 3674; Cell Signaling); mouse anti-CASPR (1:1000, catalog no. 75-001; NeuroMab); biotin-XX phalloidin (1:300, catalog no. sc-505886; Santa Cruz Biotechnology); Labeling was visualized using AF555 donkey anti-rabbit (1:300; Dianova), CF640R donkey anti-mouse (1:300; Dianova), and streptavidin AF488 (1:300; Thermo Fisher Scientific). Proteins were anchored using 0.1 mg/ml acryloyl-X (AcX, catalog no. A20770; Thermo Fisher Scientific) in PBS for 24 h at room temperature. Sections were washed, partially air-dried on an uncharged slide, incubated in gelling solution [8.6 g/100 ml sodium acrylate (Sigma-Aldrich, catalog no. 408220), 2.5 g/100 ml acrylamide (Sigma-Aldrich, catalog no. A8887), 0.1 g/100 ml N,N'-methylenebisacrylamide (Sigma-Aldrich, catalog no. M7279), 11.7 g/100 ml Sodium chloride (Sigma-Aldrich, catalog no. S6191), 0.2% TEMED accelerator solution (Sigma-Aldrich, catalog no. T9281), 0.01% 4-hydroxy-TEMPO inhibitor solution (Sigma-Aldrich, catalog no. 176141), 0.2% ammonium persulfate (Sigma Aldrich, catalog no. 248614)] for 1 h at 4 °C, and embedded in fresh gelling solution in an assembled chamber using coverslips as spacers and cover. Polymerization was performed at 37 °C for 2 h. Tissue-containing gels were trimmed, scooped off the slide, and digested in Proteinase K overnight at room temperature. Nuclei were labelled with DAPI, and gels were expanded by repeated washes in distilled water until expansion plateaued. Post-expansion specimens were imaged using a LSM900 confocal microscope (Zeiss) with an AiryScan2 detector using an LD-C Apochromat 40x/1.1 W objective. The expansion factor was calculated by measuring gels and labelled structures before and after expansion.

## Primary oligodendrocyte cell cultures

OPCs were prepared from P7 C57BL/6 J mouse brains by immunopanning[98]. Briefly, brains were digested with papain and dissociated to single-cell suspension, which was passed through two negative-selection plates coated with BSL1 to remove microglia. The remaining cell suspension was then incubated in a positive-selection plate coated with anti-PDGFRα antibodies (catalog no. sc-338; Santa Cruz Biotechnology). The attached cells were collected using trypsin and cultured on poly(L-lysine)-coated coverslips in proliferation medium containing Dulbecco's modified Eagle's medium (DMEM, catalog no. 41965; Thermo Fisher Scientific), Sato Supplement, B-27

Supplement, GlutaMAX, Trace Elements B, penicillin–streptomycin, sodium pyruvate, insulin, N-acetyl-L-cysteine, D-biotin, forskolin, ciliary neurotrophic factor (CNTF), platelet-derived growth factor (PDGF) and neurotrophin-3 (NT-3). OPCs were cultured in a differentiation medium containing DMEM (Thermo Fisher Scientific), Sato Supplement, B-27 Supplement, GlutaMAX, Trace Elements B, penicillin–streptomycin, sodium pyruvate, insulin, N-acetyl-l-cysteine, D-biotin, forskolin, CNTF and NT-3. After 7 days, when OPCs had differentiated into oligodendrocytes, they were stimulated with a medium containing sublytic[70] final concentrations (0.2 or 0.4 μg/ml) of PRF1 (catalog no. APB317Mu01; Cloud-Clone Corp.) + GZMB (catalog no. ab50114; Abcam). For experiments to inhibit ROCK-dependent actomyosin contractility, fasudil (20 μM, catalog no. orb746457; Biorbyt) was applied simultaneously. After 6 h, the cultures were fixed and analyzed by immunocytochemistry. For live imaging, oligodendrocytes were transfected with CellLight Actin-RFP (catalog no. C10502; Thermo Fisher Scientific) and imaged before and after stimulation using a Leica DMi8 microscope equipped with the DMC 2900/DFC 3000 G camera control, a stage top incubation system (Ibidi), and LAS X software (Leica).

## Live imaging of axonal transport in nerve explants

Fast axonal transport was analyzed in optic nerves and femoral quadriceps nerves of *Wt* and *PLPmut* mice hemizygous for the NMNAT2-Venus transgene as previously described[72]. Briefly, nerves were rapidly dissected into prewarmed (37 °C), pre-oxygenated Neurobasal-A medium (Gibco). Imaging of axonal transport in tissue explants was performed using a Leica DMi8 microscope equipped with the DMC 2900/DFC 3000 G camera control, a stage top incubation system (ibidi), and LAS X software (Leica). During imaging, nerves were maintained in oxygenated Neurobasal-A medium at 37 °C. Images were captured using fixed light intensity and camera exposure time settings at a rate of 2 frames per second and 60 frames over a total imaging period of 1 h. Afterwards, the nerves were fixed and analyzed by immunocytochemistry. At least 3 individual movies (often containing multiple axons) were captured for each nerve explant. Individual axons were straightened using the Straighten plugin in Fiji/ImageJ v.1.51 (National Institutes of Health). Axonal transport parameters were determined for individual axons using the DifferenceTracker[99] set of plugins using previously described parameters[72]. On average, 5 axons were analyzed for each nerve explant. For inhibition/stimulation of actomyosin contractility, nerves were incubated with medium containing DMSO (vehicle control), cytochalasin D (1 μg/ml, catalog no. C6637; Sigma-Aldrich), blebbistatin (100 μM, catalog no. B0560; Sigma-Aldrich), calyculin A (0.1 μM, catalog no. C5552; Sigma-Aldrich), or fasudil (20 μM, catalog no. orb746457; Biorbyt) and compared with nerves from the same animal fixed before and after the 1 h imaging session. Medium containing only DMSO showed no significant differences compared with pure Neurobasal-A medium.

## Fasudil treatment

Fasudil (catalog no. orb746457; Biorbyt) was prepared at 180 μg/ml within the drinking water to dose ~30 mg/kg body weight per day when given *ad libitum*. This was based on previous long-term treatment approaches in vivo which achieved efficient attenuation of brain pathology in distinct disease models[100,101]. Control mice received normal drinking water without the pharmacological inhibitor. *Wt* and *PLPmut* mice were treated for 4 weeks with daily monitoring concerning certain burden criteria and phenotypic abnormalities. The treatment started at 8 months of age when neuroinflammation and axon degeneration are ongoing in *PLPmut* mice.

## Statistics and reproducibility

All quantifications and analyses were performed by blinded investigators who were unaware of the genotype and treatment group of the

respective mice or tissue samples after concealment of groups with individual uniquely coded labels. Animals were randomly placed into experimental or control groups according to the genotyping results using a random generator (http://www.randomizer.org). For biometrical sample size estimation, G*Power v.3.1.3 was used[102]. Calculation of appropriate sample size groups was performed using an a priori power analysis by comparing the mean of 2 to 3 groups with a defined adequate power of 0.8 (1 - beta error) and an α error of 0.05. To determine the prespecified effect size d or f, previously published data were considered as comparable reference values[29]. The number of individual mice per group (number of biologically independent samples) for each experiment and the meaning of each data point are indicated in the respective figure legends. All data (except scRNA-seq) and micrographs represent at least three independent experiments with similar results. For the histological analyses, we quantified a specific cell type/structure in at least three different sections of a respective tissue and averaged the measurements into one single data point. No animals or data were excluded from the analyses. In the scRNA-seq experiment, we analyzed the brains of 3 mice for each group. Statistical analysis was performed using Prism 8 (GraphPad Software). The Shapiro–Wilk test was used to check for the normal distribution of data and the $F$ test was used to check the equality of variances to ensure that all data met the assumptions of the statistical tests used. Comparisons of two groups were performed with an unpaired Student's $t$-test (parametric comparison) or Mann-Whitney $U$-test (nonparametric comparison). For multiple comparisons, a one-way analysis of variance (ANOVA) (parametric) or Kruskal-Wallis test (nonparametric) with Tukey's post hoc test were applied and adjusted $P$ values are presented. $P < 0.05$ was considered statistically significant; exact $P$ values are provided whenever possible in the figures (for post hoc tests) and figure legends (for null hypothesis testing) with three digits and a leading zero. Only $P$ values smaller than 0.001 are shown as "$P < 0.001$".

## Reporting summary
Further information on research design is available in the Nature Portfolio Reporting Summary linked to this article.

## Data availability
The sequencing data generated in this study have been deposited in the Gene Expression Omnibus under accession number GSE224032. Source data are provided with this paper. Other data that support the findings of this study are available from the corresponding authors. Source data are provided with this paper.

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

## Acknowledgements

We thank H. Blazyca and B. Meyer for technical assistance and J. Schreiber, A. Weidner, and T. Bimmerlein for their attentive care of mice. We are grateful to D. Klein for valuable discussions. This work was supported by the German Research Foundation (grant nos. MA1053/6-2 and MA1053/7-1 to R.M. and grant no. GR5240/1-1 to J.G.), the Roman, Marga und Mareille Sobek Foundation (to R.M.), the Charitable Hertie Foundation (grant no. P1150084 to J.G.) and the Interdisciplinary Centre for Clinical Research of the University of Würzburg (grant no. A-302 to R.M.). We thank Plexxikon for generous support and providing PLX5622.

## Author contributions

J.G. and R.M. planned and oversaw all aspects of the study. J.G., T.A., Y.K., and M.H. performed and analyzed most of the experiments. S.L. performed isolation and culture of oligodendrocyte-lineage cells. V.G. and M.S. (Hannover) performed the cuprizone treatment experiment. R.A. and M.C. provided Nmnat2-Venus mice and instructions for analysis of fast axonal transport. F.I. and A.E.S. performed cell sorting and scRNA-seq. M. S. (Munich) provided equipment and substantial contributions to the conception of the work, acquisition, and interpretation of data, particularly for super-resolution microscopy and inhibition experiments. J.G. wrote the manuscript with input and substantial revisions from all authors.

## Funding

## Competing interests

The authors declare no competing interests.
