## [Peer Review File · Nature Communications]

Microglia-mediated demyelination protects against CD8+ T cell-driven axon degeneration in mice carrying PLP defectsREVIEWER COMMENTS

Reviewer #1 (Remarks to the Author):

In this manuscript, Groh et al. use a combination of Plp transgenic (PLPtg) and Plp mutant (PLPmut) mouse lines to investigate how axons with genetically perturbed myelin may be vulnerable to inflammatory damage. Additionally, the authors use electron microscopy analysis of degenerating axons, the chemical demyelinating model cuprizone, assessment of T and B cells and microglial depletion to argue that T cell secreted molecules induce cytoskeletal changes in oligodendrocytes which constrict axons and ultimately result in degeneration.

The manuscript makes a number of novel and interesting observations, including that axons are constricted in PLPmut mice at the paranode, and paradoxically (and contrary to conventional wisdom), increasing demyelination in the optic nerve with cuprizone in the PLP mutants resulted in less axonal degeneration. Unfortunately, the number of novel findings within this manuscript are limited given the relatively comprehensive existing literature on PLP mutant mice (including previous work by this laboratory). This laboratory has already identified that T cells in PLPmut (e.g., PMID 28173160) and more specifically granzyme B (e.g., PMID 20042681, PMID 22905147) in PLPtg and Plpmut contribute to neurodegeneration. Additionally, the role of microglia has also been studied in these PLPmut mice (PMID: 30565754). Arguably, the most novel contribution is the proposed mechanism by which axonal constriction due to alterations of cytoskeletal dynamics in oligodendrocytes induced by inflammatory mediators contribute to neurodegeneration. However, the causative data in favor of this mechanism is limited. Given the limited novel findings and the lack of causative cell-specific data for the proposed mechanism, in my view it does not meet the bar for publication in Nature Communications.

Major Concerns:

1. In Figure 7, the authors argue for several points of evidence of to support 'Cytotoxic effector molecules induce cytoskeletal plasticity in myelinating oligodendrocytes.' One is the increase in expression of RhoA, Cdc42 and Pak1 in oligodendrocytes. However, these genes are all upregulated in newly formed oligodendrocytes relative to OPCs and could just be secondary to increased OL differentiation following perturbations in myelin and outright demyelination observed in PLPmut mice relative to wildtype mice. Second, the authors show images of a cleaved-caspase-3 positive structure surrounded by phalloidon positive F-actin that broadly colabels with myelin, however it is unclear if this cleaved caspase-3 positive structure is indeed an axonal spheroid or merely a dying cell without colabeling of neurofilament or other markers. Lastly, the increase in F-actin in processes may be secondary to decompaction of the 2D process, thereby allowing more cytoplasm and actin filaments in, rather than reflecting a more direct action of Granzyme B and FRP1 on actin dynamics.

2. Much of the proposed mechanism hinges on the proposed activity of actin polymerization and myosin on oligodendrocyte structure and subsequent degeneration. However, the explant experiments in Figure 8D, which represents the only causative data in the paper, are not cell specific. Treatments with

Cytochalasin D and Blebbistatin would also act on the neuron directly making it challenging to conclude this cytoskeletal plasticity of oligodendrocytes results in the levels of axonal protection. These experiments also lack the control of wildtype neuron explants and whether they develop these SMI32+ constrictions over time (and if they are eliminated by Cytochalasin D and Blebbistatin).

3. In Figure 5, the authors deplete microglia using the colony stimulating factor 1 inhibitor PLX5622. It is unclear whether the increase in the number of spheroids represents reduced phagocytosis of pathological axons in the context of depleting microglia rather than increased damage. Counts of total axon number should be conducted, and retinal ganglion cell number analyzed.

4. Lines 353-356: 'Thus, efficient removal of perturbed myelin segments may allow resilience or recovery of axons at early stages of damage and argues against the current dogma that axon degeneration in myelin disease is mostly secondary to loss of myelin.' This work does support that ensheathment by perturbed myelin is detrimental to the health of axons, even relative to demyelinated axons. I agree with the authors that this is an important point. However, I would be cautious about over-concluding on this point given these results, which take place in Plp mutant mice and may be applicable for related leukodystrophies but not to other demyelinating disorders like multiple sclerosis. Indeed, it is thought that it is the long-term failure to regenerate myelin correlates strongly with neurodegeneration and disability (e.g., PMID 26891452, PMID 27671734, PMID 35152511). The authors should caveat this sentence with this point.

Minor Issues:

1. In figure 4, the role and significance of Cd11b/Cd11c/P2RY12 + microglia and what combinations of these markers actually represent in terms of microglia phenotype are never clearly stated in the text and should be added for those unfamiliar with these markers.

2. In Figure 3d, a heatmap of the top 30 differentially expressed genes in PLPmut relative to PLPtg and wildtype is shown. However, only a small number of genes are written out. This reduces its usefulness to the reader – I would recommend either showing the top 10 or so genes so it is readable and/or place the rest in a table.

3. Previous work from the authors (PMID: 30565754) demonstrated that long-term treatment of PLPmut with PLX3397 reduces the percentage of demyelinated/thinly myelinated axons as well as axonal spheroids. This previous published finding is somewhat incongruent with the argument presented that stripping myelin is broadly beneficial in protecting axons from inflammatory attack in the context of perturbed myelin. At a minimum, the authors should discuss how these results might be reconciled – is it related to T cell recruitment and activation?

4. In Supplementary Figure 2, the authors argue that CD8 cytotoxic T cells preferentially associate with damaged myelinated axons. In support of this 80% of SMI32 spheroids in myelinated axons are closely associated (<10µm) with CD8+ T cells. However, this closely parallels the percentage of myelinated axons (~80%, Figure 1h). So are the CD8+ T cells really more likely to associate with myelinated axons given they constitute the majority of the total stained axons?

Reviewer #2 (Remarks to the Author):

In previous work, the Martini group interrogated the phenotype of PLP overexpression (PLP-Tg) or PLP mutants (PLPmut) bearing human pathogenic PLP mutants on a PLPko background. In Ip et al (2006), they observed a mild demyelination phenotype in PLP-Tg, while in Groh et al (2016) they found that PLPmut showed axon degeneration, brain atrophy and impaired Rotarod performance. Notably, perturbations in both PLP-Tg and PLPmut were underlain by accumulation of CD8+ T cells. In follow up work, they implicated cognate CD8 TcR-PLP Ag interaction, and the activity of perforin/GzB as being crucial to these observations.

While in the previous studies, PLP Tg and PLPmut mice were considered separately, here the authors compared them directly (The PLPmut mice used are hPLP-R137W, as previously described). While PLPmut mice show impaired performance on functional tests as well as decreased axons, PLP-Tg show impairment in axon myelination, thus confirming previous conclusions when the two strains were separately compared to WT. Oh note, however, PLP-Tg showed an increased number of axonal spheroids than WT, indicative of an underlying axonal pathology in these mice as well. While spheroids were larger in PLPmut vs PLP-Tg, they were more heavily demyelinated in PLP-Tg.

Next, oligodendrocyte and microglial clusters were assessed for WT, PLP Tg and PLPmut mice by RNAseq. Notably, activated (AMG1-2) and proliferating (PMG) clusters were upregulated in the pathologic strains, most notably in PLP-Tg; AMG1 showed upregulation of inflammatory markers while AMG2 showed upregulation of neuroprotective ones. At the protein level, AMG1 microglia (P2RY12+) are increased in aged PLPmut, while AMG2 (P2RY12-) are enriched in PLP-Tg.

Using the cuprizone model, the authors find that PLPmut show enhanced demyelination yet less axon damage. Conversely, when PLX5622 is used to deplete microglia in PLP-Tg, one sees a reduction in abnormal demyelination yet an increase in axon damage. This seems to suggest that in the aberrant myelination may be a driver of axonal pathology, and that microglia-driven clearance of abnormal myelin may protect from axonal damage. Interestingly, treatment of ODs with perforin or granzyme elicits F-actin upregulation in vitro, and the authors posit that cytoskeletal rearrangements upon recognition of cytolytic molecules can underly changes in myelin. Formation of axon spheroids is linked to impaired axon transport of the survival factor NMNAT2.

The positive aspects of the manuscript are that the experimentation and data are of high quality. Unfortunately, these experiments, while technically quite strong, are on the whole quite descriptive - and the proposed mechanism, that the layering of perturbed myelin, and not demyelination per se, is responsible for axon degeneration -- is not really pinned down. A key limitation at this stage is the apparent lack of genetic models or blocking strategies to specifically block AMG. Some specific comments are below with the goal of helping the authors to revise this paper.

Major

Figure 4. I appreciate the logic used to arrive at the conclusions drawn, but there are some missing pieces of data that I think might be leading to over interpretation. The biggest issue, for me, is that after having gone about comparing PLPmut and PLP-Tg in parallel with WT for the first figures, the authors then compare only WT vs PLPmut (for cuprizone) and WT vs PLP-Tg for PLX depletion. The three conditions ought to be run together; importantly this is not principally for symmetry but rather to firm up the conclusions. Example: cuprizone is a model of demyelination - thus, would it not make sense to study the phenotype of PLP-Tg, in which phagocytic AMG2 monocytes are prevalent, to see whether increased phagocytosis results in less axonal damage (presumably)? Also, use of PLX in the cuprizone assay is needed to directly tie the phenotype to microglia

Another issue in Fig 4 is that PLX will deplete many types of microglia and not just the AMG subtypes.

Supplemental Figure 4C. I am not overly convinced that these GO terms are linked to axon protection or phagocytosis; there seem to be quite a few that linked to oxphos, electron transport, etc.

Supplemental Figure 5B. In the 11c/PD-1 staining, is there a possible compensation issue here? Single stain controls might assuage this concern

Minor

Fig 1B, please 2x check some of the significance values, the WT v PLP-Tg comparisons are presented as highly significant but do not seem visually evident

Supplemental 3B; as the microglia and OD were sorted prior to sequencing, can the authors clarify what ratio they were pooled at for the scRNAseq

Supplemental 3D., how were the gene functional labels assigned? Are these GO terms? It is unclear from the legend

Figure 4, the significance of CD11c/P2RY12 in identifying AMG1 vs AMG2 needs to be better explained, especially as when one looks at Fig 3, P2RY12 can mark a number of other MG types than AMG1. Is CD11c unique to AMGs? This should be elaborated upon or demonstrated looking at the scRNAseq data

Reviewer #3 (Remarks to the Author):

In the manuscript "Genetically perturbed myelin as a risk factor for neuroinflammation driven axon degeneration", Groh et al. study the interactions between T-cells, perturbed myelination and axonal degeneration. Generally, there is the assumption that the long-term loss of myelin in demyelinating

diseases like e.g. Multiple Sclerosis is causing neurodegeneration, however in this manuscript, the authors show that badly myelinated axons drive inflammation and subsequent neurodegeneration. Additionally, the manuscript offer a mechanism of how oligodendrocytes damage axons by constricting them at the paranodal domains as a reaction to the glia directed CD8+ T cell attack.

This is a well conducted study addressing an important question and therefore of high relevance for the field. The study and the experiments are well planned and conducted, however, as an not-involved reader, the manuscript is very difficult to follow and understand and there does not seem to be a red line. Especially, as the reader needs to read their previous paper about these animal models in order to understand what it is about. Maybe a small schematic figure about molecular changes in these models would help.

It is a bit confusing why the results parts starts with behavioural experiments without proper reasoning in the text. The paper starts talking about CD8 T-cells, then they go into myelin perturbations, perform scRNA-seq on microglia and then go back to CD8 T-cells and the reader does not really know why they are doing this. The scRNA-seq study (the microglia part) does somehow not fit into the story of the paper, or at least it is not becoming obvious.

Line 62ff: The authors should explain the reasoning for the PLP overexpression model, as this is crucial for the understanding of the entire manuscript. What is the pathology behind it? The reasoning for using both lines is not becoming clear throughout the manuscript

Line 64: what is TCR?

Line 142: They have sorted for O1+ oligodendrocytes but in the dataset they see very few oligodendrocytes, maybe the sorting strategy could be improved?

Response to reviewers

General remarks:

We thank all three reviewers for the insightful and helpful comments that enabled us to improve the quality and the impact of our paper. As you will see below, we responded to all the issues raised, and addressed them as far as possible with new experiments and analyses. The key changes of our revised manuscript are summarized below:

We now

- added a summarizing table of previous and novel findings in the myelin disease models (Table 1) and a short paragraph describing the main questions we tried to address with the study at the beginning of the Results part. Together, this provides background to clarify our strategy and to highlight the major novelties of our findings.
- reorganized the data and moved the previous Fig. 7 to the Supplementary Fig. 9. We show dot plots of several genes related to actomyosin GO terms that are enriched particularly in ODC from *PLPmut* mice.
- provide a co-labeling of SMI32 and cleaved caspase 3 in *PLPmut* nerves and confirmed caspase 3 activation in almost all axonal spheroids (Supplementary Fig. 9c).
- analyzed a specific marker of contractile myosin II activity (diphosphorylated MLC) using high-resolution and super-resolution fluorescence microscopy (expansion microscopy). This revealed increased myosin activity in addition to elevated F-actin levels in close association with constricted paranodal domains in *PLPmut* mice (Supplementary Fig. 9d, Fig. 7)
- added novel *in vitro* data showing that increased F-actin and pMLC levels can be induced by cytotoxic effector molecules and lead to reduction of MBP⁺ surface area in myelinating oligodendrocyte cultures (Supplementary Fig. 10).
- show that the ROCK inhibitor fasudil blocks F-actin and pMLC induction after GZMB + PRF1 treatment, as well as reduction in surface area of MBP⁺ ODC *in vitro*
- demonstrate that contractile myosin activity (pMLC) and paranodal constriction is decreased in *PLPmut/Rag1^{-/-}* mice which still show myelin decompaction (Fig. 6e, Fig. 7c, Supplementary Fig. 8c, d)
- used additional specific inhibitors to block (fasudil) or enhance (calyculin A) actomyosin contractility in explants and fasudil *in vivo* to further show that active pMLC-dependent actomyosin contractility is involved in progression of axonal spheroid formation (Fig. 9).
- analyzed nerve explants from *Wt/Nmnat2-Venus* mice which show no significant increase in spheroid numbers after the imaging process (Fig. 9b)
- provide total axon and RGC counts for PLX5622 treated mice, demonstrating that microglia depletion in *PLPtg* (but not *Wt*) mice results in aggravated axon degeneration and neuron loss (Supplementary Fig. 7d,e). Reciprocally, we also show that cuprizone treatment attenuates axon loss in *PLPmut* mice (Supplementary Fig. 7a).
- clarified the use of CD11b, CD11c, and P2RY12 in combination to discriminate between activated microglia subsets more clearly throughout the Results. We show additional immunofluorescence images to confirm this via GAL3, showing higher expression (and indicating more phagocytic activity) in AMG2 (CD11c⁺P2RY12⁻) in Fig. 4d of the revised manuscript. We also show corresponding scRNA-seq data (Fig. 3e, Supplementary Fig. 5a).
- analyzed homozygous *PLPtg* mice which show earlier removal of perturbed myelin and confirm that this also occurs without significant axon loss (Supplementary Fig. 7f)

- analyzed *Csf1* expression in *PLPmut* mice with or without cuprizone by RNAscope and found that cuprizone enhances *Csf1* mRNA expression in *Mbp*⁺ oligodendrocytes of *PLPmut* mice (Supplementary Fig. 7b)
- provide fluorescent minus one staining controls for the flow cytometric CD11c/PD-1 analysis in our revised Supplementary Fig. 5e.
- added the respective GO terms for summarizing categories to the legend of Supplementary Fig. 3d.

We incorporated text edits (changes tracked) to address all additional comments as outlined in the point-by-point responses below. Additionally, we reordered the Figures and Supplementary Figures to accommodate these changes.

Point-by-point responses:

Reviewer #1 (Remarks to the Author):

In this manuscript, Groh et al. use a combination of Plp transgenic (PLPtg) and Plp mutant (PLPmut) mouse lines to investigate how axons with genetically perturbed myelin may be vulnerable to inflammatory damage. Additionally, the authors use electron microscopy analysis of degenerating axons, the chemical demyelinating model cuprizone, assessment of T and B cells and microglial depletion to argue that T cell secreted molecules induce cytoskeletal changes in oligodendrocytes which constrict axons and ultimately result in degeneration.

The manuscript makes a number of novel and interesting observations, including that axons are constricted in PLPmut mice at the paranode, and paradoxically (and contrary to conventional wisdom), increasing demyelination in the optic nerve with cuprizone in the PLP mutants resulted in less axonal degeneration. Unfortunately, the number of novel findings within this manuscript are limited given the relatively comprehensive existing literature on PLP mutant mice (including previous work by this laboratory). This laboratory has already identified that T cells in PLPmut (e.g., PMID 28173160) and more specifically granzyme B (e.g., PMID 20042681, PMID 22905147) in PLPtg and Plpmut contribute to neurodegeneration. Additionally, the role of microglia has also been studied in these PLPmut mice (PMID: 30565754). Arguably, the most novel contribution is the proposed mechanism by which axonal constriction due to alterations of cytoskeletal dynamics in oligodendrocytes induced by inflammatory mediators contribute to neurodegeneration. However, the causative data in favor of this mechanism is limited. Given the limited novel findings and the lack of causative cell-specific data for the proposed mechanism, in my view it does not meet the bar for publication in Nature Communications.

We thank the reviewer for his interest in our work. We agree with the reviewer that immune mechanisms have been studied previously in mice with PLP defects; however, the immune-mediated mechanism that we discovered in this paper is entirely novel. Importantly, we have greatly extended the aspect of the paper that the reviewer pointed out as “novel and interesting”.

In summary, we report on the following key novel findings in mouse models of leukodystrophies:

- 1) As mentioned and acknowledged by the reviewer, we discovered that the induction of demyelination protects against axon degeneration in *PLPmut* mice.
- 2) We find that paranodal constriction mediates axon degeneration. In our revision, we extended this part of the paper with several new experiments, which now provide evidence for actomyosin contractility as the underlying mechanism (see below for details).
- 3) The role of CD8⁺ T cells and granzyme B in promoting neurodegeneration (i.e. neuronal or axonal loss) in overexpressing *PLPtg* mice has not been resolved previously. While an involvement of cytotoxic T cells in driving axonal damage and demyelination was demonstrated, it was unclear if and how this axonal damage relates to neuroaxonal loss and behavioral phenotype (we now summarize this in Table 1 of the revised manuscript). We clarified this in our present study.
- 4) The impact of targeting microglia using a CSF-1R inhibitor has been analyzed in *PLPmut* but not *PLPtg* mice previously (also see response 3 to minor concerns). We greatly extend these experiments which led to several new observations: we show an inverse relation of demyelination and neurodegeneration (including behavioral decline) when comparing the two disease models at multiple ages. We demonstrate a very different outcome of microglia depletion in *PLPtg* mice compared with our previous depletion in *PLPmut* mice. And we provide a comprehensive characterization of the interaction between ODC and microglia that can explain this distinct outcome (a beneficial phagocytic microglia population in *PLPtg* but not *PLPmut* mice).

Major Concerns:

1. In Figure 7, the authors argue for several points of evidence of to support 'Cytotoxic effector molecules induce cytoskeletal plasticity in myelinating oligodendrocytes.' One is the increase in expression of RhoA, Cdc42 and Pak1 in oligodendrocytes. However, these genes are all upregulated in newly formed oligodendrocytes relative to OPCs and could just be secondary to increased OL differentiation following perturbations in myelin and outright demyelination observed in *PLPmut* mice relative to wildtype mice. Second, the authors show images of a cleaved-caspase-3 positive structure surrounded by phalloidin positive F-actin that broadly colabels with myelin, however it is unclear if this cleaved caspase-3 positive structure is indeed an axonal spheroid or merely a dying cell without colabeling of neurofilament or other markers. Lastly, the increase in F-actin in processes may be secondary to decompaction of the 2D process, thereby allowing more cytoplasm and actin filaments in, rather than reflecting a more direct action of Granzyme B and FRP1 on actin dynamics.

Response 1: As the reviewer points out correctly, *Rhoa*, *Cdc42*, and *Pak1* are upregulated in newly formed oligodendrocytes compared to OPC. Here we used a FACS-based scRNA-seq approach to restrict the comparison of gene expression to mature (O1⁺) oligodendrocytes between the three analyzed genotypes. Nevertheless, we agree with the reviewer that experiments were required that go beyond description to determine whether immune-driven axonal spheroid formation is driven by a cytoskeleton-related process.

We have therefore performed many new and also functional experiments and analyses to study the process in more detail.

- 1) We show dot plots of several genes related to actomyosin GO terms that are enriched particularly in ODC from *PLPmut* mice (Supplementary Fig. 9a). We provide a co-labeling of SMI32 and cleaved caspase3 in *PLPmut* nerves and confirmed caspase 3 activation in almost all (~98%) axonal spheroids. (Supplementary Fig. 9c).

- 2) Moreover, we analyzed a specific marker of contractile myosin II activity (diphosphorylated MLC) using high-resolution and super-resolution fluorescence microscopy (expansion microscopy) and observed increased myosin activity in addition to elevated F-actin levels in close association with constricted paranodal domains in *PLPmut* mice (Supplementary Fig. 9d, Fig. 7).
- 3) In addition, novel *in vitro* data show that increased F-actin and pMLC levels can be induced by cytotoxic effector molecules and lead to constriction of MBP⁺ surface area in myelinating oligodendrocyte cultures (Supplementary Fig. 10). Moreover, we now show that the ROCK inhibitor fasudil blocks F-actin and pMLC induction after GZMB + PRF1 treatment, as well as reduction in surface area of MBP⁺ ODC *in vitro* (also see response 2).
- 4) Furthermore, we demonstrate that contractile myosin activity (pMLC) and paranodal constriction is decreased in *PLPmut/Rag1^{-/-}* mice which still show myelin decompaction (Fig. 6e, Fig. 7c, Supplementary Fig. 8c, d).
- 5) Most importantly, we also used additional specific inhibitors to block (fasudil) or enhance (calyculin A) actomyosin contractility in explants and fasudil *in vivo*. These experiments support a role for active pMLC-dependent actomyosin contractility in the progression of axonal spheroid formation (also see response 2).

We cannot exclude that some of the cytoskeletal changes that we observe are a consequence of myelin perturbation (such as decompaction). However, with these new functional experiments, we now provide strong evidence for immune-mediated actomyosin contractility as an underlying mechanism in focal axon degeneration.

2. Much of the proposed mechanism hinges on the proposed activity of actin polymerization and myosin on oligodendrocyte structure and subsequent degeneration. However, the explant experiments in Figure 8D, which represents the only causative data in the paper, are not cell specific. Treatments with Cytochalasin D and Blebbistatin would also act on the neuron directly making it challenging to conclude this cytoskeletal plasticity of oligodendrocytes results in the levels of axonal protection. These experiments also lack the control of wildtype neuron explants and whether they develop these SMI32⁺ constrictions over time (and if they are eliminated by Cytochalasin D and Blebbistatin).

Response 2: We thank the reviewer for raising an important point related to the cell specificity of our inhibition experiments and agree that we presently cannot formally exclude that axon-intrinsic cytoskeletal changes might contribute to the observed reduction in axon diameter. We think that an unequivocal proof of principle to demonstrate this *in vivo* would take several years of additional work, since we would have to generate a conditional (oligodendrocyte-specific) knockout of actomyosin components. This would likely have to be inducible since cytoskeletal dynamics are very important during developmental myelination and would then have to be introduced into the *PLPmut* mouse line. Such an approach is beyond the scope of our present work and represents an important follow-up project. Instead, we therefore performed several additional experiments to study the role of cytoskeletal dynamics and more specifically actomyosin constriction in axon degeneration. We used additional specific inhibitors to block (fasudil) or enhance (calyculin A) actomyosin contractility in optic nerve explants to show that active pMLC-dependent actomyosin contractility is involved in progression of axonal spheroid formation (Fig. 9). While cytochalasin D, blebbistatin, and fasudil inhibit MLC activation and spheroid formation, calyculin A increases pMLC levels and axonal spheroid formation in optic nerve explants. Moreover, we now demonstrate *in vitro* that fasudil inhibits the induction of F-actin and pMLC levels, as well as the reduction in myelin sheet surface area of oligodendrocytes (Supplementary Fig. 10f). Finally, we show that treatment of myelin mutants

with fasudil attenuates ongoing axonal spheroid formation *in vivo* (Fig. 9e). We followed the suggestion of the reviewer and analyzed nerve explants from *Wt/Nmnat2-Venus* mice which show no significant increase in spheroid numbers after the imaging process (Fig. 9b). This further indicates that immune-driven and actomyosin-dependent spheroid formation in explants requires perturbed oligodendrocytes, at least in the here investigated timeframe. A direct mode of action on the neuron would be difficult to explain since the membrane periodic skeleton maintains axonal integrity and its disruption via cytoskeletal inhibitors accelerates beading and fragmentation *in vitro*^{1,2} (contrary to the protective effects in our explant experiments). Combined with our above-described observations (see response 1), we think that it is rather unlikely that neuronal cytoskeletal changes drive axonal spheroid formation. However, we reworded the manuscript and discuss that we can presently not fully exclude this possibility.

3. In Figure 5, the authors deplete microglia using the colony stimulating factor 1 inhibitor PLX5622. It is unclear whether the increase in the number of spheroids represents reduced phagocytosis of pathological axons in the context of depleting microglia rather than increased damage. Counts of total axon number should be conducted, and retinal ganglion cell number analyzed.

Response 3: The reviewer raises a valid point, and we now provide total axon and RGC counts for PLX5622 treated mice, demonstrating that microglia depletion in *PLPtg* (but not *Wt*) mice results in aggravated axon degeneration and neuron loss (Supplementary Fig. 7d,e). Reciprocally, we also show that cuprizone treatment attenuates axon loss in *PLPmut* mice (Supplementary Fig. 7a).

4. Lines 353-356: 'Thus, efficient removal of perturbed myelin segments may allow resilience or recovery of axons at early stages of damage and argues against the current dogma that axon degeneration in myelin disease is mostly secondary to loss of myelin.' This work does support that ensheathment by perturbed myelin is detrimental to the health of axons, even relative to demyelinated axons. I agree with the authors that this is an important point. However, I would be cautious about over-concluding on this point given these results, which take place in *Plp* mutant mice and may be applicable for related leukodystrophies but not to other demyelinating disorders like multiple sclerosis. Indeed, it is thought that it is the long-term failure to regenerate myelin correlates strongly with neurodegeneration and disability (e.g., PMID 26891452, PMID 27671734, PMID 35152511). The authors should caveat this sentence with this point.

Response 4: We appreciate the point raised and want to clarify that we did not intend to state that demyelination is always beneficial or that remyelination is not beneficial. Interestingly, in MS and its models, similar findings have recently been obtained³. Thus, persistence of dysfunctional myelin in general might be detrimental to axonal integrity when targeted by immune cells. We added the suggested citations and discuss that chronic failure in remyelination can also be detrimental to neuronal integrity (3rd paragraph of Discussion). In case of myelin mutants, it is interesting to speculate about the benefit of remyelination, since regenerated myelin will still be perturbed due to the gene defect.

Minor Issues:

1. In figure 4, the role and significance of Cd11b/Cd11c/P2RY12 + microglia and what combinations of these markers actually represent in terms of microglia phenotype are never clearly stated in the text and should be added for those unfamiliar with these markers.

Response 1: We clarified the use of CD11b, CD11c, and P2RY12 in combination to discriminate between activated microglia subsets more clearly throughout the Results. CD11c (encoded by *Itgax*) is an established DAM marker⁴ specific to AMG and P2RY12 levels discriminate AMG1 from AMG2 (also see response 4 to minor concerns of reviewer 2). We furthermore show additional immunofluorescence images to confirm this via GAL3, showing higher expression (and indicating more phagocytic activity) in AMG2 (CD11c⁺P2RY12⁻) in Fig. 4d of the revised manuscript.

2. In Figure 3d, a heatmap of the top 30 differentially expressed genes in PLPmut relative to PLPtg and wildtype is shown. However, only a small number of genes are written out. This reduces its usefulness to the reader – I would recommend either showing the top 10 or so genes so it is readable and/or place the rest in a table.

Response 2: We think that labelling the top10 differentially expressed genes (DEGs) would make the heatmap less informative since many of the most significant DEGs are related to ribosomal genes. We therefore would like to keep the present display and indicate some of the more interesting genes without overcrowding it. Supplementary table 1 lists all significant DEGs.

3. Previous work from the authors (PMID: 30565754) demonstrated that long-term treatment of PLPmut with PLX3397 reduces the percentage of demyelinated/thinly myelinated axons as well as axonal spheroids. This previous published finding is somewhat incongruent with the argument presented that stripping myelin is broadly beneficial in protecting axons from inflammatory attack in the context of perturbed myelin. At a minimum, the authors should discuss how these results might be reconciled – is it related to T cell recruitment and activation?

Response 3: The reviewer is correct that our previous work showed that microglia depletion with a CSF-1R inhibitor attenuated CD8⁺ T cell recruitment and axonal damage in *PLPmut* mice. In the present work we show that the same approach leads to a detrimental outcome in *PLPtg* mice. We propose that this is related to the additional beneficial function of microglia regarding removal of perturbed myelin in the latter disease model. While microglia in both models promote CD8⁺ T cell recruitment and activation initially (AMG1), microglia in *PLPmut* mice fail to efficiently phagocytose myelin, resulting in a chronic detrimental disease progression. In contrast, microglia in *PLPtg* mice additionally strip myelin efficiently (removing immune target and detrimental axon-glia interaction) and may provide tolerance (AMG2), resulting in axonal survival and resolution of inflammation. Therefore, the different outcome upon microglia depletion is related to distinct neural-immune interactions and represents a major novel finding. We expanded our discussion on this.

4. In Supplementary Figure 2, the authors argue that CD8 cytotoxic T cells preferentially associate with damaged myelinated axons. In support of this 80% of SMI32 spheroids in myelinated axons are closely associated (<10µm) with CD8+ T cells. However, this closely parallels the percentage of myelinated axons (~80%, Figure 1h). So are the CD8+ T cells really more likely to associate with myelinated axons given they constitute the majority of the total stained axons?

Response 4: There is most likely a mix-up with the models. We added an additional label into the figure to make this point clearer. In Supplementary Fig. 2, we demonstrate this association for *PLPtg* but not *PLPmut* mice. In *PLPtg* mice, only 20% of the optic nerve axons are still myelinated at this age (Figure 1h). Nevertheless, damaged SMI32⁺ axons associated with CD8⁺ T cells are to 80% myelinated, showing a strong preference of the T cells towards damaged axons that are still myelinated. In other words: strongly demyelinated regions in *PLPtg* mice show lower numbers of CD8⁺ T cells (likely because the immune target is gone). We analyzed this only in *PLPtg* mice due to the scarce and diffuse demyelination in *PLPmut* mice.

Reviewer #2 (Remarks to the Author):

In previous work, the Martini group interrogated the phenotype of PLP overexpression (PLP-Tg) or PLP mutants (PLPmut) bearing human pathogenic PLP mutants on a PLPko background. In Ip et al (2006), they observed a mild demyelination phenotype in PLP-Tg, while in Groh et al (2016) they found that PLPmut showed axon degeneration, brain atrophy and impaired Rotarod performance. Notably, perturbations in both PLP-Tg and PLPmut were underlaid by accumulation of CD8⁺ T cells. In follow up work, they implicated cognate CD8 TcR-PLP Ag interaction, and the activity of perforin/GzB as being crucial to these observations.

While in the previous studies, PLP Tg and PLPmut mice were considered separately, here the authors compared them directly (The PLPmut mice used are hPLP-R137W, as previously described). While PLPmut mice show impaired performance on functional tests as well as decreased axons, PLP-Tg show impairment in axon myelination, thus confirming previous conclusions when the two strains were separately compared to WT. Oh note, however, PLP-Tg showed an increased number of axonal spheroids than WT, indicative of an underlying axonal pathology in these mice as well. While spheroids were larger in PLPmut vs PLP-Tg, they were more heavily demyelinated in PLP-Tg.

Next, oligodendrocyte and microglial clusters were assessed for WT, PLP Tg and PLPmut mice by RNAseq. Notably, activated (AMG1-2) and proliferating (PMG) clusters were upregulated in the pathologic strains, most notably in PLP-Tg; AMG1 showed upregulation of inflammatory markers while AMG2 showed upregulation of neuroprotective ones. At the protein level, AMG1 microglia (P2RY12⁺) are increased in aged PLPmut, while AMG2 (P2RY12⁻) are enriched in PLP-Tg.

Using the cuprizone model, the authors find that PLPmut show enhanced demyelination yet less axon damage. Conversely, when PLX5622 is used to deplete microglia in PLP-Tg, one sees a reduction in abnormal demyelination yet an increase in axon damage. This seems to suggest that in the aberrant myelination may be a driver of axonal pathology, and that microglia-driven clearance of abnormal myelin may protect from axonal damage. Interestingly, treatment of ODs with perforin or granzyme elicits F-actin upregulation in vitro, and the authors posit that cytoskeletal rearrangements upon recognition of cytolytic molecules can underly changes in myelin. Formation of axon spheroids is linked to impaired axon transport of the survival factor NMNAT2.

The positive aspects of the manuscript are that the experimentation and data are of high quality. Unfortunately, these experiments, while technically quite strong, are on the whole quite descriptive - and the proposed mechanism, that the layering of perturbed myelin, and not

demyelination per se, is responsible for axon degeneration -- is not really pinned down. A key limitation at this stage is the apparent lack of genetic models or blocking strategies to specifically block AMG. Some specific comments are below with the goal of helping the authors to revise this paper.

We thank the reviewer for his thorough evaluation of our present and previous work and for appreciating the quality of our data. We followed the suggestions of the reviewer and revised the manuscript as detailed below.

Major

1. Figure 4. I appreciate the logic used to arrive at the conclusions drawn, but there are some missing pieces of data that I think might be leading to over interpretation. The biggest issue, for me, is that after having gone about comparing PLPmut and PLP-Tg in parallel with WT for the first figures, the authors then compare only WT vs PLPmut (for cuprizone) and WT vs PLP-Tg for PLX depletion. The three conditions ought to be run together; importantly this is not principally for symmetry but rather to firm up the conclusions. Example: cuprizone is a model of demyelination - thus, would it not make sense to study the phenotype of PLP-Tg, in which phagocytic AMG2 monocytes are prevalent, to see whether increased phagocytosis results in less axonal damage (presumably)? Also, use of PLX in the cuprizone assay is needed to directly tie the phenotype to microglia

Response 1: The reviewer raises an important issue regarding our approaches to modulate microglial myelin phagocytosis and possibly missing pieces of data. We have previously already used a CSF-1R inhibitor to deplete microglia in *PLPmut* mice and interestingly observed a beneficial outcome (see also our response 3 to reviewer 1). This different outcome upon depletion in the two disease models is related to distinct neural-immune interactions (more beneficial phagocytosis of perturbed myelin in *PLPtg*) and represents a major novel finding. We furthermore think it would not be a feasible approach to enhance demyelination in *PLPtg* mice with cuprizone for the following reasons: cuprizone is given to mice at an age of 6 to 10 weeks and is much less efficient at later ages⁵. At this early age, *PLPtg* mice do not show significant demyelination (Fig. 2d,e, Supplementary Fig. 7f), little axonal damage (Supplementary Fig. 1c), and no axon degeneration yet (Supplementary Fig. 7f). Even at later ages, axonal loss remains very mild in *PLPtg* mice. Thus, there would be no axon degeneration that could be attenuated by earlier removal of perturbed myelin with cuprizone. We tested this hypothesis by analyzing homozygous *PLPtg* mice which show earlier removal of perturbed myelin and confirm that this also occurs without significant axon loss (Supplementary Fig. 7f). Previous work performed with a combination of cuprizone + PLX3397 has shown that CSF-1-related phagocytosis by microglia is necessary for demyelination after cuprizone feeding⁶. To further characterize if cuprizone might affect demyelination-promoting neural-immune interactions, we analyzed *Csf1* expression in *PLPmut* mice with or without cuprizone by RNAscope. Indeed, cuprizone enhanced *Csf1* mRNA expression in *Mbp*⁺ oligodendrocytes of *PLPmut* mice (Supplementary Fig. 7b). Moreover, cuprizone-treated *PLPmut* mice displayed microglia containing myelin fragments and lysosomal storage material resembling AMG2 observed in *PLPtg* mice (Supplementary Fig. 7c), supporting our previous finding of GAL3 induction (Fig. 4f). We furthermore provide total axon counts for *PLPmut* mice after cuprizone and *PLPtg* mice after PLX5622 treatment to firm up our conclusions (Supplementary Fig. 7a,d,e). This shows that cuprizone indeed attenuates the axon loss occurring early in *PLPmut* mice, while PLX5622 results in aggravated axon degeneration and neuron loss in *PLPtg* mice (also see response 3 to reviewer 1).

Another issue in Fig 4 is that PLX will deplete many types of microglia and not just the AMG subtypes.

While it is true that PLX5622 treatment will non-specifically deplete most microglia (homeostatic ones even more effectively), it is an established approach to test microglial function in different conditions⁷. Considering the feasibility of using this approach compared with long-lasting conditional genetic deletion experiments of the here identified subset markers, we think this approach is valid and allows our present conclusions. AMG1 and AMG2 are the two microglia subsets specifically enriched in our models of myelin disease. Since depletion of microglia in *Wt* mice has no effects on the analyzed parameters of CNS integrity, we think that the effects observed in *PLP*^{Tg} mice are related to the depletion of activated microglia (particularly AMG2 in *PLP*^{Tg} mice).

2. Supplemental Figure 4C. I am not overly convinced that these GO terms are linked to axon protection or phagocytosis; there seem to be quite a few that linked to oxphos, electron transport, etc.

Response 2: It is true that oxidative phosphorylation is one of the top GO terms enriched in AMG2. However, oxidative phosphorylation is known to be enhanced in tolerogenic phagocytes that protect from oxidative damage and resolve inflammation⁸. Moreover, we only showed the top 20 significant hits in the bar graph but “Detoxification of ROS”, “Microglia pathogen phagocytosis pathway”, and “Immunoregulatory interactions between a Lymphoid and a Non-Lymphoid cell” also were found in the top 100 hits during the metascape analysis. The enriched genes for AMG2 shown in Fig. 3e are also known from previous studies to regulate phagocytosis (*Tyrobp*, *Cst7*, *Lyz2*, *Fabp5*, *Lgals3*)^{4,9} and axon protection (*Fth1*, *Ftl1*, *Cox8a*, *Spp1*, *Igf1*)^{10,11}.

3. Supplemental Figure 5B. In the 11c/PD-1 staining, is there a possible compensation issue here? Single stain controls might assuage this concern

Response 3: The reviewer raises a valid concern, since microglial autofluorescence sometimes makes it difficult to discriminate compensation issues from real signal. We provide fluorescent minus one staining controls for the flow cytometric CD11c/PD-1 analysis in our revised Supplementary Fig. 5e showing that there is not a compensation problem.

Minor

1. Fig 1B, please 2x check some of the significance values, the WT v PLP-Tg comparisons are presented as highly significant but do not seem visually evident

Response 1: We thank the reviewer for bringing this to our attention. There was a mistake when indicating the correct *P* values in the mentioned graph. We corrected it and double checked all statistical tests for possible mistakes.

2. Supplemental 3B; as the microglia and OD were sorted prior to sequencing, can the authors clarify what ratio they were pooled at for the scRNAseq

Response 2: Microglia and ODC were pooled at a 1 to 1 ratio during the sorting (we sorted 15,000 cells each as indicated in the Methods section) but ODC appeared much more vulnerable during the further processing after sorting. This is why we ended up with a much

higher number of analyzed microglia. We are in the process of improving this with revised protocols and describe this limitation in the Methods section.

3. Supplemental 3D., how were the gene functional labels assigned? Are these GO terms? It is unclear from the legend

Response 3: These are categories assigned based on the literature and summarize/simplify GO terms of the respective genes (proteostasis: cytoplasmic translation, lysosome; inflammation: microglial activation involved in immune response, complement binding, lysozyme activity; protection from phagocytosis: negative regulation of apoptotic process, negative regulation of Fc-gamma receptor signaling pathway involved in phagocytosis; promotion of phagocytosis: microglial cell activation, complement activation, cholesterol homeostasis). We now added the respective GO terms to the legend of Supplementary Fig. 3d.

4. Figure 4, the significance of CD11c/P2RY12 in identifying AMG1 vs AMG2 needs to be better explained, especially as when one looks at Fig 3, P2RY12 can mark a number of other MG types than AMG1. Is CD11c unique to AMGs? This should be elaborated upon or demonstrated looking at the scRNAseq data

Response 4: CD11c (encoded by *Itgax*) is an established marker for disease-associated microglia⁴ and in the present models specific to the AMG subsets (also see response 1 to minor concerns of reviewer 1). P2RY12 levels can discriminate AMG1 from AMG2. We provide further details to explain this at multiple instances and show corresponding scRNA-seq data (Fig. 3e, Supplementary Fig. 5a). We furthermore show additional immunofluorescence images to confirm this via GAL3, showing higher expression (and indicating more phagocytic activity) in AMG2 (CD11c⁺P2RY12⁺) in Fig. 4d of the revised manuscript.

Reviewer #3 (Remarks to the Author):

In the manuscript “Genetically perturbed myelin as a risk factor for neuroinflammation driven axon degeneration”, Groh et al. study the interactions between T-cells, perturbed myelination and axonal degeneration. Generally, there is the assumption that the long-term loss of myelin in demyelinating diseases like e.g. Multiple Sclerosis is causing neurodegeneration, however in this manuscript, the authors show that badly myelinated axons drive inflammation and subsequent neurodegeneration. Additionally, the manuscript offer a mechanism of how oligodendrocytes damage axons by constricting them at the paranodal domains as a reaction to the glia directed CD8+ T cell attack. This is a well conducted study addressing an important question and therefore of high relevance for the field. The study and the experiments are well planned and conducted, however, as an not-involved reader, the manuscript is very difficult to follow and understand and there does not seem to be a red line. Especially, as the reader needs to read their previous paper about these animal models in order to understand what it is about. Maybe a small schematic figure about molecular changes in these models would help.

We thank the reviewer for his evaluation and for appreciating the relevance and execution of our experiments. To make the manuscript easier to understand, we added a summarizing table detailing previous and novel findings regarding similarities and differences between the two myelin disease models.

1. It is a bit confusing why the results parts starts with behavioural experiments without proper reasoning in the text. The paper starts talking about CD8 T-cells, then they go into myelin perturbations, perform scRNA-seq on microglia and then go back to CD8 T-cells and the reader does not really know why they are doing this. The scRNA-seq study (the microglia part) does somehow not fit into the story of the paper, or at least it is not becoming obvious.

Line 62ff: The authors should explain the reasoning for the PLP overexpression model, as this is crucial for the understanding of the entire manuscript. What is the pathology behind it? The reasoning for using both lines is not becoming clear throughout the manuscript

Response 1: In addition to adding a summarizing table of previous and novel findings in the disease models (Table 1), we now added a short paragraph describing the main questions we tried to address with the study at the beginning of the Results part. Together, this introduces the reason why we performed behavioral experiments first and provides background regarding the impact of CD8⁺ T cells. Moreover, it provides background on why we chose to specifically compare these two disease models. The characterization of oligodendrocyte-microglia interactions can explain the differences in microglia-mediated demyelination between the models and the distinct neurodegenerative outcome despite similar initial damage (see also our responses to reviewers 1 and 2). We tried to make these points clearer in the revised manuscript.

2. Line 64: what is TCR?

Response 2: T cell receptor. We introduced the abbreviation.

3. Line 142: They have sorted for O1+ oligodendrocytes but in the dataset they see very few oligodendrocytes, maybe the sorting strategy could be improved?

Response 3: Yes, ODC are far more vulnerable and seem to lose viability during/after the sorting process. We are in the process of improving this with revised protocols and describe this limitation in the Methods section (see also response 2 to minor concerns of reviewer 2).

References

- 1 Unsain, N. *et al.* Remodeling of the Actin/Spectrin Membrane-associated Periodic Skeleton, Growth Cone Collapse and F-Actin Decrease during Axonal Degeneration. *Sci Rep* **8**, 3007 (2018).
- 2 Datar, A. *et al.* The Roles of Microtubules and Membrane Tension in Axonal Beading, Retraction, and Atrophy. *Biophys J* **117**, 880-891 (2019).
- 3 Schäffner, E. *et al.* Myelin insulation as a risk factor for axonal degeneration in autoimmune demyelinating disease. *Nat Neurosci* (2023).
- 4 Deczkowska, A. *et al.* Disease-Associated Microglia: A Universal Immune Sensor of Neurodegeneration. *Cell* **173**, 1073-1081 (2018).
- 5 Gingele, S. *et al.* Delayed Demyelination and Impaired Remyelination in Aged Mice in the Cuprizone Model. *Cells* **9** (2020).
- 6 Marzan, D. E. *et al.* Activated microglia drive demyelination via CSF1R signaling. *Glia* **69**, 1583-1604 (2021).
- 7 Green, K. N., Crapser, J. D. & Hohsfield, L. A. To Kill a Microglia: A Case for CSF1R Inhibitors. *Trends Immunol* **41**, 771-784 (2020).
- 8 Gonzalez, M. A. *et al.* Phagocytosis increases an oxidative metabolic and immune suppressive signature in tumor macrophages. *J Exp Med* **220** (2023).

- 9 Rotshenker, S. The role of Galectin-3/MAC-2 in the activation of the innate-immune function of phagocytosis in microglia in injury and disease. *J Mol Neurosci* **39**, 99-103 (2009).
- 10 Mukherjee, C. *et al.* Oligodendrocytes Provide Antioxidant Defense Function for Neurons by Secreting Ferritin Heavy Chain. *Cell Metab* **32**, 259-272 e210 (2020).
- 11 Li, S. & Jakobs, T. C. Secreted phosphoprotein 1 slows neurodegeneration and rescues visual function in mouse models of aging and glaucoma. *Cell Rep* **41**, 111880 (2022).

REVIEWER COMMENTS

Reviewer #1 (Remarks to the Author):

In this revised manuscript the authors have thoughtfully and thoroughly addressed all my comments and concerns. I have no further comments and consider the manuscript highly suitable for Nature Communications in its current form.

Reviewer #2 (Remarks to the Author):

The manuscript has evidently been revised with numerous new pieces of data. The good faith attempt to address concerns regarding Figure 8 (raised by another reviewer) are appreciated. However, several of my initial concerns still stand:

Major Point 1.1 (regarding Figure 5):

After reading the authors' response and re-reviewing the manuscript (and returning to the response), I think I have finally "decoded" the approach. In essence, for PLPmut, cuprizone is used as a "therapy" to induce myelin loss, thereby discouraging the formation of axonal spheroids down the road. By contrast, in PLP-Tg, PLX depletion is used to prevent demyelination which thus causes spheroid formation. Based on this, I agree that the symmetrical controls are not needed.

That being said, the text is extremely confusing to follow (another Reviewer made a similar remark). I urge the authors to make their approach and rationale MUCH more apparent to the reader. Perhaps (only a suggestion): "Based on [prior], we wanted to test the hypothesis that microglia-dependent demyelination can prevent axonal spheroid formation in pathological conditions. Thus we used two complementary approaches, one in which we forced microglial-dependent demyelination in PLPmut using cuprizone, and another in which we ablated microglia in PLP-Tg"....

Major Point 1.2 (regarding lack of AMG targeting):

More needs to be done to assuage this concern. I can accept that targeting AMG at this time is not feasible. However, the only evidence of AMG2 modulation is based on gal-3 positivity. This is quite thin - can they at a minimum co-stain for AMG1/HMG markers (P2RY12?) to show that they are negative for these while gal-3+? At least in the cuprizone system - if it will take 6 months to get a result with PLX depletion. The argument that AMG is specifically altered in the PLP lines is a bit weak because while true, the data from 3A show that these are still a minority of the microglial cells overall.

Major Point 2: (re: SUpP Fig 4C):

I would be more convinced of the involvement of axon protection/phagocytosis if NES plots for relevant terms were presented. Also, in the graphs, are the P-values adjusted (ie Benjamini-Hochberg correction or similar)?

Other critiques in my first review (mostly minor) have been satisfactorily addressed.

Additional issues:

8C. What is the significance of the control (DMSO) in increasing spheroids so dramatically? Is this a natural result of the culture conditions or is this caused by DMSO itself? As someone unfamiliar with assay (and perhaps reflective of the broader readership of Nat Comm), some explanation would be useful.

Reviewer #3 (Remarks to the Author):

The authors have addressed all points and the manuscript is now a lot clearer to understand and should be accepted.

Response to reviewers

General remarks:

We thank all three reviewers for evaluating our revised manuscript. Reviewers 1 and 3 had no further concerns or comments. We responded to all the remaining issues raised by reviewer 2 and addressed them with new data and text edits (all changes highlighted).

Point-by-point responses:

Reviewer #1 (Remarks to the Author):

In this revised manuscript the authors have thoughtfully and thoroughly addressed all my comments and concerns. I have no further comments and consider the manuscript highly suitable for Nature Communications in its current form.

Reviewer #2 (Remarks to the Author):

The manuscript has evidently been revised with numerous new pieces of data. The good faith attempt to address concerns regarding Figure 8 (raised by another reviewer) are appreciated. However, several of my initial concerns still stand:

Major Point 1.1 (regarding Figure 5):

After reading the authors' response and re-reviewing the manuscript (and returning to the response), I think I have finally "decoded" the approach. In essence, for PLPmut, cuprizone is used as a "therapy" to induce myelin loss, thereby discouraging the formation of axonal spheroids down the road. By contrast, in PLP-Tg, PLX depletion is used to prevent demyelination which thus causes spheroid formation. Based on this, I agree that the symmetrical controls are not needed.

That being said, the text is extremely confusing to follow (another Reviewer made a similar remark). I urge the authors to make their approach and rationale MUCH more apparent to the reader. Perhaps (only a suggestion): "Based on [prior], we wanted to test the hypothesis that microglia-dependent demyelination can prevent axonal spheroid formation in pathological conditions. Thus we used two complementary approaches, one in which we forced microglial-dependent demyelination in PLPmut using cuprizone, and another in which we ablated microglia in PLP-Tg"....

Response 1.1: We followed the suggestion of the reviewer and revised the respective paragraph in the Results part to make our rationale clearer.

Major Point 1.2 (regarding lack of AMG targeting):

More needs to be done to assuage this concern. I can accept that targeting AMG at this time is not feasible. However, the only evidence of AMG2 modulation is based on gal-3 positivity. This is quite thin - can they at a minimum co-stain for AMG1/HMG markers (P2RY12?) to show that they are negative for these while gal-3+? At least in the cuprizone system - if it will take 6 months to get a result with PLX depletion. The argument that AMG is specifically altered in the PLP lines is a bit weak because while true, the data from 3A show that these are still a minority of the microglial cells overall.

Response 1.2: As suggested by the reviewer, we now provide additional quantitative data regarding the co-expression of HMG (CD11c/P2RY12⁺) and AMG1 (CD11c⁺/P2RY12⁺) markers by GAL3⁺ microglia in our revised Fig. 5e,f (replacing the previous quantifications) after both cuprizone treatment in *PLPmut* mice and PLX5622 depletion in *PLPtg* mice. In line with our electron microscopy-based results on myelin phagocytosis (Fig. 5b,d, Supplementary Fig. 7c), this confirms that cuprizone treatment increases the numbers of AMG2 in the white matter of *PLPmut* mice, whereas PLX5622 treatment decreases the numbers of AMG2 (among other clusters/states) in *PLPtg* mice. While AMG represent a minority among the different microglia clusters, they are strongly enriched in the myelin disease models (disease-associated/specific) and one should consider that the whole brain (including gray matter) was used for isolation of microglial cells for scRNA-seq. Moreover, the involvement of other clusters in myelin phagocytosis is difficult to explain based on marker characterization and the fact that microglial depletion in *Wt* mice (showing low numbers of AMG) has no effect on myelin integrity.

Major Point 2: (re: SUp Fig 4C):

I would be more convinced of the involvement of axon protection/phagocytosis if NES plots for relevant terms were presented. Also, in the graphs, are the P-values adjusted (ie Benjamini-Hochberg correction or similar)?

Response 2: We added the respective enrichment bar graph of the top 100 hits for our recent metascape analysis of AMG2 (and AMG1) genes to Supplementary Fig. 4c (replacing the previous top 20 hits) which include, for example: "Microglia pathogen phagocytosis pathway", "Phagosome", and "Detoxification of ROS". Regarding adjustment for multiple comparisons, metascape provides details here: <https://metascape.org/blog/?p=122>. While the enrichment bar graphs generated by metascape show non-adjusted *P*-values (FDR makes use of some assumptions that cannot be validated for enrichment analysis), metascape also provides *Q*-values (BH-corrected). The above-mentioned terms were significantly enriched when considering the *Q*-value ($Q < 0.05$). Moreover, we added additional references supporting a role for some of the enriched genes in phagocytosis and axon protection to the manuscript.

Other critiques in my first review (mostly minor) have been satisfactorily addressed.

Additional issues:

8C. What is the significance of the control (DMSO) in increasing spheroids so dramatically? Is this a natural result of the culture conditions or is this caused by DMSO itself? As someone

unfamiliar with assay (and perhaps reflective of the broader readership of Nat Comm), some explanation would be useful.

Response 3: DMSO itself does not affect ongoing axonal spheroid formation in optic nerve explants of *PLPmut* mice at the concentrations used in our study. It is used as a vehicle control since the applied drugs (blebbistatin, cytochalasin D, calyculin A) were dissolved in DMSO. The increased number of axonal spheroids after culture of the explants is a result of the ongoing pathological process in the myelin disease model. We explained this in the revised manuscript (Results and Methods parts).

Reviewer #3 (Remarks to the Author):

The authors have addressed all points and the manuscript is now a lot clearer to understand and should be accepted.

REVIEWERS' COMMENTS

Reviewer #2 (Remarks to the Author):

The authors have satisfactorily addressed my concerns. One comment though, in Fig 5E, it is not clear from which experimental condition the sample images are derived.

Response to reviewers

Point-by-point responses:

Reviewer #2 (Remarks to the Author):

The authors have satisfactorily addressed my concerns. One comment though, in Fig 5E, it is not clear from which experimental condition the sample images are derived.

Response: We have added the requested information about the experimental condition to the representative images shown in Fig. 5e.